# Weakly Supervised Representation Learning with Sparse Perturbations

**Kartik Ahuja**[*]    **Jason Hartford**[*]    **Yoshua Bengio**[†*]

## Abstract

The theory of representation learning aims to build methods that provably invert the data generating process with minimal domain knowledge or any source of supervision. Most prior approaches require strong distributional assumptions on the latent variables and weak supervision (auxiliary information such as timestamps) to provide provable identification guarantees. In this work, we show that if one has weak supervision from observations generated by sparse perturbations of the latent variables–e.g. images in a reinforcement learning environment where actions move individual sprites–identification is achievable under unknown continuous latent distributions. We show that if the perturbations are applied only on mutually exclusive blocks of latents, we identify the latents up to those blocks. We also show that if these perturbation blocks overlap, we identify latents up to the smallest blocks shared across perturbations. Consequently, if there are blocks that intersect in one latent variable only, then such latents are identified up to permutation and scaling. We propose a natural estimation procedure based on this theory and illustrate it on low-dimensional synthetic and image-based experiments.

## 1 Introduction

If you are reading this paper on a computer, press one of the arrow keys... all the text you are reading jumps as the screen refreshes in response to your action. Now imagine you were playing a video game like Atari's Space Invaders—the same keystroke would cause a small sprite at the bottom of your screen to move in response. These actions induce changes in pixels that are very different, but in both cases, the visual feedback in response to our actions indicates the presence of some object on the screen—a virtual paper and a virtual spacecraft, respectively—with properties that we can manipulate. Our keystrokes induce sparse changes to a program's state, and these changes are reflected on the screen, albeit not necessarily in a correspondingly sparse way (e.g., most pixels change when scrolling). Similarly, many of our interactions with the real world induce sparse changes to the underlying causal factors of our environment: lift a coffee cup and the cup moves, but not the rest of the objects on your desk; turn your head laterally, and the coordinates of all the objects in the room shift, but only in the horizontal direction. These examples hint at the main question we aim to answer in this paper: if we know that actions have sparse effects on the latent factors of our system, can we use that knowledge as weak supervision to help disentangle these latent factors from pixel-level data?

Self– and weakly-supervised learning approaches have made phenomenal progress in the last few years, with large-scale systems like GPT-3 (Brown et al., 2020) offering large improvements on all natural language benchmarks, and CLIP (Radford et al., 2021) outperforming state-of-the-art supervised models from six years ago (Szegedy et al., 2016) on the ImageNet challenge (Deng et al., 2009) without using any of the labels.

---

[*]Mila - Quebec AI Institute, Université de Montréal.  [†]CIFAR Fellow.
Correspondence to: kartik.ahuja@mila.quebec.

36th Conference on Neural Information Processing Systems (NeurIPS 2022).

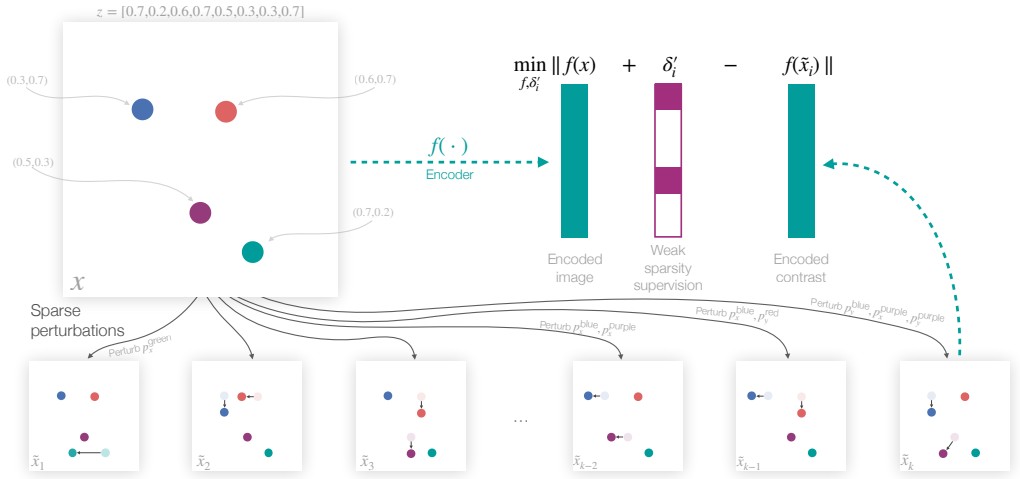

Figure 1: Ball agent interaction environment. Different frames show the effect of sparse perturbations.

Yet, despite these advances, these systems are still far from human reasoning abilities and often fail on out-of-distribution examples (Geirhos et al., 2020). To robustly generalize out of distribution, we need models that can infer the causal mechanisms that relate latent variables (Schölkopf et al., 2021; Schölkopf and von Kügelgen, 2022) because these mechanisms are invariant under distribution shift. The field of causal inference has developed theory and methods to infer causal mechanisms from data (Pearl, 2009; Peters et al., 2017), but these methods assume access to high-level abstract features, instead of low-level signal data such as video, text and images. We need representation learning methods that reliably recover these abstract features if we are to bridge the gap between causal inference and deep learning.

This is a challenging task because the problem of inferring latent variables is not identified with independent and identically distributed (IID) data (Hyvärinen and Pajunen, 1999; Locatello et al., 2019), even in the limit of an infinite number of such IID examples. However, there has been significant recent progress in developing representation learning approaches that provably recover latent factors $Z$ (e.g., object positions, object colors, etc.) underlying complex data $X$ (e.g. image), where $X \leftarrow g(Z)$, by going beyond the IID setting and using observations of $X$ along with minimal domain knowledge and supervision (Hyvarinen and Morioka, 2016, 2017; Locatello et al., 2020; Khemakhem et al., 2020a). These works establish provable identification of latents by leveraging strong structural assumptions such as independence conditional on auxiliary information (e.g., timestamps). In this work, we aim to relax these distributional assumptions on the latent variables to achieve identification for arbitrary continuous latent distributions. Instead of distributional assumptions, we assume access to data generated under sparse perturbations that change only a few latent variables at a time as a source of weak supervision. Figure 1 illustrates our working example of this assumption: a simple environment where an agent's actions perturb the coordinates of a few balls at a time. Our main contributions are summarized as follows.

- We show that sparse perturbations that impact one latent at a time are sufficient to learn the latents (up to permutation and scaling) that follow any unknown continuous distribution.

- Next, we consider more general settings, where perturbations impact one block of latent variables at a time. In the setting where blocks do not overlap, we recover the latents up to an affine transformation of these blocks.

- Further, we show that when perturbation blocks overlap, we get stronger identification. In this setting, we prove identification up to affine transformation of the smallest intersecting block. Consequently, if there are blocks that intersect in one latent variable only, then such latents are identified up to permutation and scaling.

- We leverage these results to propose a natural estimation procedure and experimentally illustrate the theoretical claims on low-dimensional synthetic and high-dimensional image-based data.

## 2 Related works

Many of the works on provable identification of representations trace their roots to non-linear ICA (Hyvärinen and Pajunen, 1999). Hyvarinen and Morioka (2016, 2017) were the first to use auxiliary information in the form of timestamps and additional structure on the latent evolution to achieve provable identification. Since then, these works have been generalized in many exciting ways. Khemakhem et al. (2020a) assume independence of latents conditional on auxiliary information, and several of these assumptions were further relaxed by Khemakhem et al. (2020b).

Our work builds on the machinery developed by Ahuja et al. (2022). Ahuja et al. show that if we know the mechanisms that drive the evolution of latents, then the latents are identified up to equivariances of these mechanisms. However, the authors leave the question of achieving identification without such knowledge open. Here we consider a class of mechanisms where an agent's actions impact the latents through unknown perturbations. We show how to achieve identification by exploiting the sparsity in the perturbations. This class of perturbations was first leveraged to prove identification by Locatello et al. (2020). However, Locatello et al. assume that the latents are independent, whereas we make no assumptions on the distribution other than continuity. Lachapelle et al. (2022) also use sparse interventions on the latents to strengthen the identification guarantees in Khemakhem et al. (2020a) for conditional exponential distributions. However, the form of sparsity that is leveraged in their work is different from ours. In our work, we assume that the vector of changes in the latents is sparse, i.e., some components change and the rest of the components do not change. In Lachapelle et al. (2022), all the components of the latents change post interventions but the graphical model capturing the interaction of interventions (described as random variables) and the latents is sparse. Klindt et al. (2021) also use sparsity in time-series settings to attain identification. Klindt et al. (2021) enforce soft $\ell_1$ norm driven sparsity in the vector of changes in latents by assuming that latents evolve independently under a Laplace distribution but do not require access to data under interventions.

Yao et al. (2021) and Lippe et al. (2022) model the latent evolution as a structural causal model unrolled in time. Yao et al. exploit non-stationarity and sufficient variability dictated by the auxiliary information to provide identification guarantees. Lippe et al. exploit causal interventions on the latents to provide identification guarantees but require the knowledge of intervention targets and assume an invariant causal model describing the relations between any adjacent time frames. In concurrent work, Brehmer et al. (2022) leverage data generated under causal interventions as a source of weak supervision and prove identification for structural causal models that are diffeomorphic transforms of exogenous noise. Our work also connects to an insightful line of work on multi-view ICA (Gresele et al., 2020), which proves identification under independent latents, in the following sense. We can interpret the data under different perturbations as different views of the same underlying latent. In addition, some recent papers explain the success of self-supervised contrastive learning Zimmermann et al. (2021); Von Kügelgen et al. (2021) through the lens of identification of representations.

Above, we focused on provable representation identification, which is central to this work. We now give a brief overview of empirical works on disentanglement which have shown success on some benchmark tasks, but do not theoretically characterize conditions for successful disentanglement. Many variations of variational autoencoders (VAE) were developed over the years to achieve disentanglement. $\beta$-VAE (Higgins et al., 2016) uses a hyperparameter in front of the KL regularizer to make the learned latent independent. Factor VAE (Kim and Mnih, 2018) proposes an adversarial training-based approach, where the discriminator encourages the learned representation to have independent components. Annealed $\beta$-VAE (Burgess et al., 2018) proposes to progressively increase the capacity of bottleneck $\beta$ to enforce independence one component at a time. Ideas based on disentanglement have been also used in reinforcement learning; Higgins et al. (2017), Dittadi et al. (2020), and Miladinović et al. (2019) are some of the representative works in the area. Locatello et al. (2019) showed that most of the above methods could often fail to disentangle in the absence of supervision or inductive biases. As a result, there has been a surge in the interest in building approaches that achieve provable representation identification. Lastly, there is a line of work, which does not focus on disentanglement or representation identification, but has shown the benefits of sparsity based inductive biases – sparse changes in latents over time (Goyal et al., 2019) or sparse interactions between latents (Goyal et al., 2021) – under distribution shifts.

# 3 Latent identification under sparse perturbations

**Data Generation Process** We start by describing the data generation process used for the rest of the work. There are two classes of variables we consider – a) unobserved latent variables $Z \in \mathcal{Z} \subseteq \mathbb{R}^d$ and b) observed variables $X \in \mathcal{X} \subseteq \mathbb{R}^n$. The latent variables $Z$ are sampled from a distribution $\mathbb{P}_Z$ and then transformed by a map $g : \mathbb{R}^d \to \mathbb{R}^n$, where $g$ is injective and analytic[2], to generate $X$. We write this as follows

$$z \sim \mathbb{P}_Z \qquad x \leftarrow g(z) \tag{1}$$

where $z$ and $x$ are realizations of the random variables $Z$ and $X$ respectively. It is impossible to invert $g$ just from the realizations of $X$ (Hyvärinen and Pajunen, 1999; Locatello et al., 2019). Most work has gone into understanding how structure of latents $Z$ and auxiliary information in the form of timestamps or labels play a role in solving the above problem. In this work, we depart from these assumptions and instead investigate the role of data generated under perturbations of latents to achieve identification. Define the set of perturbations as $\mathcal{I} = \{1, \cdots, m\}$ and the corresponding perturbation vectors as $\Delta = \{\delta_1, \cdots, \delta_m\}$, where $\delta_i$ is the $i^{th}$ perturbation. Each latent $z$ is sampled from an arbitrary and unknown distribution $\mathbb{P}_Z$. The *same set of unknown perturbations* in $\Delta$ are applied to each $z$ to generate $m$ perturbed latents $\{\tilde{z}_k\}_{k=1}^m$ per sampled $z$ and the corresponding observed vectors $\{\tilde{x}_k\}_{k=1}^m$. Each of these latents are transformed by the map $g$ and we observe $(x, \tilde{x}_1, \cdots, \tilde{x}_m)$. Our goal is to use these observations and estimate the underlying latents. We summarize this data generation process (DGP) in the following assumption.

**Assumption 1.** *The DGP follows*

$$z \sim \mathbb{P}_Z, x \leftarrow g(z) \qquad \tilde{z}_k \leftarrow z + \delta_k, \forall k \in \mathcal{I} \qquad \tilde{x}_k \leftarrow g(\tilde{z}_k), \forall k \in \mathcal{I} \tag{2}$$

*where $g$ is injective and analytic, and $Z$ is a continuous random vector with full support over $\mathbb{R}^d$.* [3]

The above DGP is very close to the DGP in Locatello et al. (2020) except we do not require latent dimensions to be mutually independent. To better understand the above DGP, let us turn to some examples. Consider a setting where an agent is interacting with an environment containing several balls (See Figure 1). The latent $z$ captures the properties of the objects; for example, in Figure 1, $z$ just captures the positions of each ball, but in general it could include more properties such as velocity, shape, color, etc.. The agent perturbs the objects in the scene by $\delta_k$, which can modify a single property associated with one object or multiple properties from one or more objects depending on how the agent acts. Note that when the agent perturbs a latent, it can lead to downstream effects. For instance, if the agent moves a ball to the edge of the table, the ball falls in subsequent frames. For this work, we only consider the observations just before and after the perturbation and not the downstream effects. In the Appendix (Section A.2.5), we explain these downstream effects using structural causal models. We also explain the connection between the perturbations in equation (24) and causal interventions leveraged in Brehmer et al. (2022); Lachapelle et al. (2022). The above example is typical of a reinforcement learning environment, other examples include natural videos with sparse changes (e.g., MPI3D data (Gondal et al., 2019)).

In the above DGP in equation (24), we assumed that for each scene $x$ there are multiple perturbations. It is possible to extend our results to settings where we perturb each scene only once, given a sufficiently diverse set of perturbations, i.e., for a small neighborhood of a scene around $x$, each scene in the neighbourhood receives a different perturbation. We compare these two approaches experimentally.

**Learning objective** The learner's objective is to use the observed samples $(x, \tilde{x}_1, \cdots, \tilde{x}_m)$ generated by the DGP in Assumption 1 and learn an encoder $f : \mathbb{R}^n \to \mathbb{R}^d$ that inverts the function $g$ and recovers the true latents. For each observed sample $(x, \tilde{x}_1, \cdots, \tilde{x}_m)$, the learner compares all the pairs $(x, \tilde{x}_k)$ pre- and post-perturbation. For every unknown perturbation $\delta_k$ used in the DGP in equation (24), the learner guesses the perturbation $\delta_k'$ and enforces that the latents predicted by the encoder for $x$ and $\tilde{x}_k$ are consistent with the guess. We write this as $\forall (x, \tilde{x}_1, \cdots, \tilde{x}_m)$ generated by DGP in (24)

$$f(\tilde{x}_k) = f(x) + \delta_k'. \tag{3}$$

---

[2]A *analytic* function, $g$, is an infinitely differentiable function such that for all $z'$ in its domain, the Taylor series evaluated at $z'$ converges pointwise to $g(z')$

[3]The assumption on the support of $Z$ can be relaxed.

We denote the set of guessed perturbations as $\Delta' = \{\delta'_1, \cdots, \delta'_m\}$, where $\delta'_i$ is the guess for perturbation $\delta_i$. We can turn the above identity into a mean square error loss given as

$$\min_{f, \Delta'} \mathbb{E}\left[\left(f(\tilde{x}_k) - f(x) - \delta'_k\right)^2\right] \qquad (4)$$

where the expectation is taken over observed samples generated by the DGP in (24) and the minimization is over all the possible maps $f$ and perturbation guesses in the set $\Delta'$. Note that a trivial solution to the above problem is an encoder that maps everything to zero, and all guesses equal zero. In the next section, we get rid of these trivial solutions by imposing an additional requirement that the span of the set $\Delta'$ is $\mathbb{R}^d$. It is worth pointing out that we do not restrict the set of $f$'s to injective maps in theory and experiments. We denote the latent estimated by the encoder for a point $x$ as $\hat{z} = f(x)$. It is related to the true latent as follows $\hat{z} = f \circ g(z) = a(z)$, where $a$ is some function that relates true $z$ to estimated $\hat{z}$. In the next section, we show that if perturbations are diverse, then $a$ is an affine transform. Further, we show that if perturbations are sparse, then $a$ takes an even simpler form.

## 3.1 Sparse perturbations

We first show that it is possible to identify the true latents up to an affine transformation without any sparsity assumptions. Later, we leverage sparsity to strengthen identification guarantees.

**Assumption 2.** *The dimension of the span of the perturbations in (24) is $d$, i.e.,* $\dim\left(\mathsf{span}(\Delta)\right) = d$.

The above assumption implies that the perturbations are diverse. We now state a regularity condition on the function $a$.

**Assumption 3.** *$a$ is an analytic function. For each component $i \in \{1, \cdots, d\}$ of $a(z)$ and each component $j \in \{1, \cdots, d\}$ of $z$, define the set $\mathcal{S}^{ij} = \{\theta \mid \nabla_j a_i(z+b) = \nabla_j a_i(z) + \nabla_j^2 a_i(\theta)b, z \in \mathbb{R}^d\}$, where $b$ is a fixed vector in $\mathbb{R}^d$. Each set $\mathcal{S}^{ij}$ has a non-zero Lebesgue measure in $\mathbb{R}^d$.*

If we restrict the encoder $f$ to be analytic, then $a$ is analytic since $g$ is also analytic, thus satisfying the first part of the above assumption. The second part of the above assumption can be understood as follows: suppose we have a scalar valued function $h : \mathbb{R} \to \mathbb{R}$ that is differentiable. If we expand $h(u + v)$ around $h(u)$, by the mean value theorem we get $h(u + v) = h(u) + h'(\epsilon)v$, where $\epsilon \in [u, u + v]$. If we vary $u$ to take all the values in $\mathbb{R}$, then $\epsilon$ also varies. The above assumption states that the set of $\epsilon's$ has a non-zero Lebesgue measure. Under the above assumptions, we show that an encoder that solves equation (3) identifies true latents up to an affine transform, i.e., $\hat{z} = Az + c$, where $A \in \mathbb{R}^{d \times d}$ is a matrix and $c \in \mathbb{R}^d$ is an offset.

**Proposition 1.** *If Assumptions 1, 2, and 3 hold, then the encoder that solves equation (3) (with $\Delta'$ s.t. $\dim\left(\mathsf{span}(\Delta')\right) = d$) identifies true latents up to an invertible affine transform, i.e. $\hat{z} = Az + c$, where $A \in \mathbb{R}^{d \times d}$ is an invertible matrix and $c \in \mathbb{R}^d$ is an offset.*

The proof of above proposition follows the proof technique from Ahuja et al. (2022), for further details refer to the Appendix (Section A.1). We interpret the above result in the context of the agent interacting with balls (as shown in Figure 1), where the latent vector $z$ captures the $x$ and $y$ coordinates of the $n_{\mathsf{balls}}$. Under each perturbation, the balls move along the vector dictated by the perturbation. If there are at least $2n_{\mathsf{balls}}$ perturbations, then the latents estimated by the learned encoder are guaranteed to be an affine transformation of the actual positions of the balls.

### 3.1.1 Non-overlapping perturbations

In Proposition 1, we showed affine identification guarantees for the DGP from Assumption 1. We now explore identification when perturbations are one-sparse, i.e., one latent changes at a time.

**Assumption 4.** *The perturbations in $\Delta$ are one-sparse, i.e., each $\delta_i \in \Delta$ has one non-zero component.*

Next, we show that under one-sparse perturbations, the latents estimated identify true latents up to permutation and scaling.

**Theorem 1.** *If Assumptions 1-4 hold and the number of perturbations per example equals the latent dimension, $m = d$,* [4] *then the encoder that solves equation (3) (with $\Delta^{'}$ as one-sparse and $\dim\left(\text{span}(\Delta^{'})\right) = d$) identifies true latents up to permutation and scaling, i.e. $\hat{z} = \Pi\Lambda z + c$, where $\Lambda \in \mathbb{R}^{d \times d}$ is an invertible diagonal matrix, $\Pi \in \mathbb{R}^{d \times d}$ is a permutation matrix and $c$ is an offset.*

For the proof of above theorem, refer to Section A.1 in the Appendix. The theorem does not require that learner knows either the identity or amount each component changed. However, the learner has to use one-sparse perturbations as guesses. Suppose the learner does not know that the actual perturbations are one-sparse and instead uses guesses that are $p$-sparse, i.e., $p$ latents change at one time. In such a case, the $\hat{z}$ and true $z$ are related to each other through a permutation and block diagonal matrix, i.e., we can replace $\Lambda$ in the above result to be a block diagonal matrix instead of a diagonal matrix (see the Appendix for details). In the context of the ball agent interaction environment from Figure 1, the above result states that provided the agent interacts with one coordinate of each ball at a time, it is possible to learn the position of each ball up to scaling errors.

We now consider a natural extension of the setting above, where the perturbations simultaneously operate on blocks of latents. In the ball agent interaction environment, this can lead to multiple scenarios – i) the agent interacts with one ball at a time but perturbs both coordinates simultaneously, ii) the agent interacts with several balls simultaneously.

Consider a perturbation $\delta_i \in \Delta$ (from equation (24)). We define the block of latents that is impacted under perturbation $\delta_i \in \Delta$ as $\{j \mid \delta_i^j \neq 0, j \in \{1, \cdots, d\}\}$, where $\delta_i^j$ is the $j^{th}$ component of $\delta_i$. We group the perturbations in $\mathcal{I}$ based on the block they act upon, i.e. perturbations in the same group act on the same block of latents. Define the set of the groups corresponding to perturbations in $\mathcal{I}$ as $\mathcal{G}_{\mathcal{I}}$. Define the set of corresponding blocks as $\mathcal{B}_{\mathcal{I}} = \{\mathcal{B}_1, \cdots, \mathcal{B}_g\}$, where $\mathcal{B}_k$ is the block impacted by perturbations in group $k$. If $\mathcal{B}_{\mathcal{I}}$ partitions the set of latent components indexed $\{1, \cdots, d\}$, then it implies all the distinct blocks are non-overlapping. We formally define this below.

**Definition 1.** *Blockwise and non-overlapping perturbations. If the the set of blocks $\mathcal{B}_{\mathcal{I}}$ corresponding to perturbations $\mathcal{I}$ form a partition of $\{1, \cdots, d\}$, then $\mathcal{I}$ is said to be blockwise and non-overlapping. Formally stated, any two distinct $\mathcal{B}_i, \mathcal{B}_j \in \mathcal{B}_{\mathcal{I}}$ do not intersect, i.e., $\mathcal{B}_i \cap \mathcal{B}_j = \emptyset$, and $\cup_i \mathcal{B}_i = \{1, \cdots, d\}$.*

From the above definition it follows that two perturbations either act on the same block or completely different blocks with no overlapping variables.

**Assumption 5.** *The perturbations $\mathcal{I}$ (used in equation (24)) are blockwise and non-overlapping (see Definition 1). Each perturbation in $\mathcal{I}$ is $p$-sparse, i.e., it impacts blocks of length $p$ ($p \leq d$) at a time.*

**Assumption 6.** *The learner knows the group label for each perturbation $i \in \mathcal{I}$. Therefore, any two perturbations in $\Delta^{'}$ associated with same group in $\mathcal{G}_{\mathcal{I}}$ impact the same block of latents.*

We illustrate the above Assumptions 5, 6 in the following example. Consider the ball agent interaction environment (Figure 1). $z = [z_{1x}, z_{1y}, \cdots, z_{n_{\text{balls}}x}, z_{n_{\text{balls}}y}]$ is the vector of positions of all balls, where $z_{ix/y}$ is the $x/y$ coordinate of ball $i$. If the agent randomly perturbs ball $i$, then it changes the block $(z_{ix}, z_{iy})$. We would call such a system 2-sparse. All the perturbations on ball $i$ are in one group. Since the agent knows the group of the perturbation, it does not know the ball index but it knows whenever we interact with the same ball.

**Definition 2.** *If the latent variables recovered $\hat{z} = \Pi\tilde{\Lambda}z + c$, where $\Pi$ is a permutation matrix and $\tilde{\Lambda}$ is a block-diagonal matrix, then the latent variables are said to be recovered up to permutations and block-diagonal transforms.*

In the theorem that follows, we show that under the assumptions made in this section, we achieve identification up to permutations and block-diagonal transforms with invertible $p \times p$ blocks.

**Theorem 2.** *If Assumptions 1-3, 5, 6 hold, then the encoder that solves equation (3) (where $\Delta^{'}$ is $p$-sparse, $\dim\left(\text{span}(\Delta^{'})\right) = d$) identifies true latents up to permutation and block-diagonal transforms, i.e. $f(x) = \hat{z} = \Pi\tilde{\Lambda}z + c$, where $\tilde{\Lambda} \in \mathbb{R}^{d \times d}$ is an invertible block-diagonal matrix with blocks of size $p \times p$, $\Pi \in \mathbb{R}^{d \times d}$ is a permutation matrix and $c \in \mathbb{R}^d$ is an offset.*

---

[4]We can relax this condition to $m \geq d$, refer to the Appendix for details.

For the proof of the above theorem, refer to Section A.1 in the Appendix. From the above theorem, we gather that the learner can separate the perturbed blocks. However, the latent dimensions within the block are linearly entangled. In the ball agent interaction with 2-sparse perturbations, the above theorem implies that the agent can separate each ball out but not their respective $x$ and $y$ coordinates. In the above theorem, we require the learner to know the group of each intervention (Assumption 6). In the Appendix Section A.2.2, we relax Assumption 6 and show that we can continue to achieve identification up to permutation and block diagonal transforms. However, we need a computationally expensive procedure that searches over subsets of latent dimensions to identify the dimensions impacted under the current intervention.

We briefly compare with Von Kügelgen et al. (2021), where the authors also establish block identification guarantees. In Von Kügelgen et al. (2021), the latent vector is divided into two parts – the content block and the style block. Across augmentations, style is varied and content is fixed. Von Kügelgen et al. leverage this invariance of the content across augmentations to learn the content block and not the style block. To summarize, invariance of content across different views is the key signal that is used to achieve identification. In our case, the perturbations act on different blocks of latents. In contrast to Von Kügelgen et al., we leverage sparsity of changes, i.e., we exploit both the varying part and the invariant part to identify all the distinct blocks and not just the content block.

### 3.1.2 Overlapping perturbations

In the previous section, we assumed that the blocks across different perturbations are non-overlapping. This section relaxes this assumption and allows the perturbation blocks to overlap. We start with a motivating example to show how overlapping perturbations can lead to stronger identification.

Consider the agent interacting with two balls, where $z = [z_{1x}, z_{1y}, z_{2x}, z_{2y}]$ describes the coordinates of the two balls. The agent perturbs the first ball and then perturbs the second ball. For the purpose of this example, assume that these perturbations satisfy the assumptions in Theorem 2. We obtain that the estimated position of each ball $\hat{z}_{ix/y}$ is linearly entangled w.r.t the true $x$ and $y$ coordinates. For the first ball we get $\hat{z}_{1x} = a_1 z_{1x} + a_2 z_{1y} + a_3$. We also have the agent perturb the $x$ coordinates of the first and second ball together and then it does the same with the $y$ coordinates. We apply Theorem 2 and obtain that the estimated $x$ coordinates of each ball are linearly entangled. We write this as $\hat{z}_{1x} = b_1 z_{1x} + b_2 z_{2x} + b_3$. We take a difference of the two relations for $\hat{z}_{1x}$ to get

$$(a_1 - b_1)z_{1x} + a_2 z_{1y} - b_2 z_{2x} + a_3 - b_3 = 0 \tag{5}$$

Since the above has to hold for all $z_{1x}, z_{1y}, z_{2x}$, we get $a_1 = b_1$, $a_2 = 0$, $b_2 = 0$ and $a_3 = b_3$. Thus $\hat{z}_{1x} = a_1 z_{1x} + a_3$. Similarly, we can disentangle the rest of the balls.

We take the insights from the above example and generalize them below. Let us suppose that from the set of perturbations $\mathcal{I}$ we can construct at least two distinct subsets $\mathcal{I}_1$ and $\mathcal{I}_2$ such that both subsets form a blockwise non-overlapping perturbation (see Definition 1). Perturbations in $\mathcal{I}_1$ ($\mathcal{I}_2$) partition $\{1, \cdots, d\}$ into blocks $\mathcal{B}_{\mathcal{I}_1}$ ($\mathcal{B}_{\mathcal{I}_2}$) respectively. It follows that there exists at least two blocks $\mathcal{B}^1 \in \mathcal{B}_{\mathcal{I}_1}$ and $\mathcal{B}^2 \in \mathcal{B}_{\mathcal{I}_2}$ such that $\mathcal{B}^1 \cap \mathcal{B}^2 \neq \emptyset$. From Theorem 2, we know that we can identify latents in block $\mathcal{B}^1$ and $\mathcal{B}^2$ up to affine transforms. In the next theorem, we show that we can identify latents in each of the blocks $\mathcal{B}^1 \cap \mathcal{B}^2$, $\mathcal{B}^1 \setminus \mathcal{B}^2$, $\mathcal{B}^2 \setminus \mathcal{B}^1$ up to affine transforms.

**Assumption 7.** *Each perturbation in $\mathcal{I}$ is $p$-sparse. The perturbations in each group span a $p$-dimensional space, i.e., $\forall q \in \mathcal{G}_{\mathcal{I}}$, $\dim\left(\mathsf{span}\left(\{\delta_i\}_{i \in q}\right)\right) = p$. There exist at least two distinct subsets of perturbations $\mathcal{I}_1 \subseteq \mathcal{I}$ and $\mathcal{I}_2 \subseteq \mathcal{I}$ that are both blockwise and non-overlapping.*

**Theorem 3.** *Suppose Assumptions 1, 3, 6 and 7 hold. Consider the subsets $\mathcal{I}_1$ and $\mathcal{I}_2$ that satisfy Assumption 7. For every pair of blocks, $\mathcal{B}^1 \in \mathcal{B}_{\mathcal{I}_1}$ and $\mathcal{B}^2 \in \mathcal{B}_{\mathcal{I}_2}$, the encoder that solves equation (3) (where $\Delta'$ is $p$-sparse, $\dim\left(\mathsf{span}(\Delta')\right) = d$) identifies latents in each of the blocks $\mathcal{B}^1 \cap \mathcal{B}^2$, $\mathcal{B}^1 \setminus \mathcal{B}^2$, $\mathcal{B}^2 \setminus \mathcal{B}^1$ up to invertible affine transforms.*

For the proof of the above theorem, refer to Section A.1 in the Appendix. From the above theorem, it follows that if blocks overlap at one latent only, then all such latents are identified up to permutation and scaling. We now construct an example to show the identification of all the latents under overlapping perturbations. Suppose we have a 4 dimensional latent. The set of all contiguous blocks of length 2 is given as follows $\{\{1, 2\}, \{2, 3\}, \{3, 4\}, \{4, 1\}\}$. Different 2-sparse perturbations impact

these blocks. Observe that every component between 1 to 4 gets to be the first element of a block exactly once and the last element of the block exactly once. As a result, each latent gets to be the only element at the intersection of two blocks. We apply Theorem 3 to this case and get that all the latents are identified up to permutation and scaling. We generalize this example below.

**Assumption 8.** *$\mathcal{B}_\mathcal{I}$ is a set of all the contiguous blocks of length $p$, where $p < d$. The perturbations in each block span a $p$ dimensional space. Further, also assume that $d \bmod p = 0$.*

In the above assumption, we construct $d$ contiguous blocks of length $p$. The construction ensures that each index in $\{1, \cdots, d\}$ forms the first element of exactly one block and last element of exactly one block. In Theorem 1 (Locatello et al., 2019) and in Theorem 5 (Lachapelle et al., 2022) a similar assumption is made that requires exactly one latent is at the intersection of multiple blocks. In the next theorem, we show that under the above assumption, we achieve identification up to permutation and scaling.

**Theorem 4.** *Suppose Assumptions 1, 3, 6 and 8 hold, then the encoder that solves the identity in equation (3) (where $\Delta'$ is $p$-sparse, $\dim\left(\mathsf{span}\left(\Delta'\right)\right) = d$) identifies true latents up to permutations and scaling, i.e., $\hat{z} = \Pi\Lambda z + c$, where $\Pi \in \mathbb{R}^{d \times d}$ matrix and $\Lambda \in \mathbb{R}^{d \times d}$ is a diagonal matrix.*

For the proof of the above theorem, refer to Section A.1 in the Appendix. The total number of perturbations required in the above theorem is $p \times d$. If we plug $p = 1$, we recover Theorem 1 as a special case. The above result highlights that if the block lengths are larger, then we need to scale the number of perturbations accordingly by the same factor to achieve identification up to permutation and scaling. We assumed a special class of perturbations operating on contiguous blocks. In general, the total number of distinct blocks can be up to $\binom{d}{p}$. Suppose $s$ distinct random blocks of length $p$ are selected for perturbations. As $s$ grows, we reach a point where each latent component is at the intersection of two blocks from different sets of blockwise non-overlapping perturbations. At that point, we identify all latents up to permutation and scaling.

**Extensions**    In the discussion so far, we made some assumptions for ease of exposition. In the appendix, we describe how to relax them. In the DGP in Assumption 1, the perturbations used are deterministic. In Section A.2.3, we extend the DGP in Assumption 1 to incorporate stochastic perturbations. Specifically, instead of $\tilde{z} \leftarrow z + \delta$ we consider a DGP where $\tilde{z} \leftarrow z + \delta + n$, where $n$ is the noise vector added to the perturbation $\delta$. We show that the key results presented in the paper extend provided the noise vector $n$ follow the same sparsity pattern as $\delta$. We also present experiments for the same model in Section A.3. In the DGP in Assumption 1, the perturbations used are independent of the value of $z$. Instead of $\tilde{z} \leftarrow z + \delta$ we consider a DGP given as $\tilde{z} \leftarrow z + m(z)$, where $m(\cdot)$ is a general non-linear perturbation map. In Section A.2.4, we show that the key results presented in the paper extend to this setting with non-linear perturbation mechanisms

## 4   Experiments

**Data generation processes**    We conducted two sets of experiments – low-dimensional synthetic and image-based inputs – that follow the DGP in equation (24). In the low-dimensional synthetic experiments we experimented with two choices for $\mathbb{P}_Z$ a) uniform distribution with independent latents, b) normal distribution with latents that are blockwise independent (with block length $d/2$). We used an invertible multi-layer perceptron (MLP) (with 2 hidden layers) from Zimmermann et al. (2021) for $g$. We evaluated for latent dimensions $d \in \{6, 10, 20\}$. The training and test data size was 10000 and 5000 respectively. For the image-based experiments we used PyGame (Shinners, 2011)'s rendering engine for $g$ and generated $64 \times 64$ pixel images that look like those shown in Figure 1. The coordinates of each ball, $z_i$, were drawn independently from a uniform distribution, $z_i \sim \mathcal{U}(0.1, 0.9)$. We varied the number of balls from 2 ($d = 4$) to 4 ($d = 8$). For these experiments, there was no fixed-size training set; instead the images are generated online and we trained to convergence. Because these problems are high dimensional, we only sampled a single perturbation for each image.

**Loss function, architecture, evaluation metrics**    In all the experiments we optimized equation (4) with square error loss. The encoder $f$ was an MLP with two hidden layers of size 100 for the low-dimensional synthetic experiments and a ResNet-18 (He et al., 2015) for the image-based experiments. Further training details such as the optimizers used, hyperparameters etc. are in the

Table 1: Comparing MCC and BMCC for non-overlapping perturbations. The number of perturbations applied for each example is given in parenthesis

| $d$ | $p_Z$ | MCC C-wise ($d$) | MCC C-wise (1) | BMCC B-wise ($d$) | BMCC B-wise (1) |
|---|---|---|---|---|---|
| 6 | Normal | $0.99 \pm 0.00$ | $0.99 \pm 0.00$ | $0.99 \pm 0.00$ | $0.99 \pm 0.01$ |
| 10 | Normal | $0.99 \pm 0.00$ | $0.99 \pm 0.01$ | $0.99 \pm 0.00$ | $0.91 \pm 0.02$ |
| 20 | Normal | $0.99 \pm 0.00$ | $0.88 \pm 0.03$ | $0.99 \pm 0.00$ | $0.90 \pm 0.01$ |
| 6 | Uniform | $0.99 \pm 0.00$ | $0.99 \pm 0.00$ | $0.99 \pm 0.00$ | $0.96 \pm 0.04$ |
| 10 | Uniform | $0.99 \pm 0.00$ | $0.99 \pm 0.01$ | $0.99 \pm 0.00$ | $0.81 \pm 0.05$ |
| 20 | Uniform | $0.99 \pm 0.00$ | $0.82 \pm 0.02$ | $0.85 \pm 0.08$ | $0.51 \pm 0.04$ |

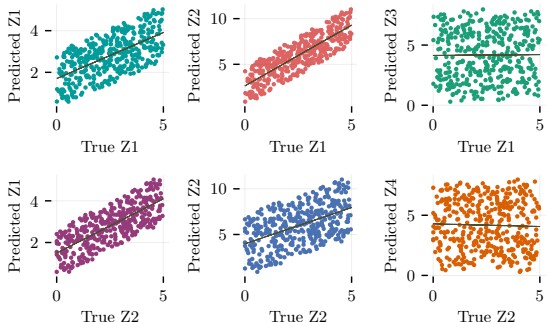

Figure 2: Illustrating blockwise dependence ($d = 10$).

Table 2: MCC for B-wise (overlap).

| $d$ | Distribution | MCC |
|---|---|---|
| 6 | Normal | $0.95 \pm 0.01$ |
| 10 | Normal | $0.96 \pm 0.01$ |
| 20 | Normal | $0.99 \pm 0.01$ |
| 6 | Uniform | $0.86 \pm 0.03$ |
| 10 | Uniform | $0.88 \pm 0.03$ |
| 20 | Uniform | $0.81 \pm 0.03$ |

Appendix (Section A.3). We used the mean correlation coefficient (MCC) (Hyvarinen and Morioka, 2016) to verify the claims in Theorems 1 and 4. If MCC equals one, then the estimated latents identify true latents up to permutation and scaling. We extend MCC to blockwise MCC (BMCC) to verify the claims in Theorem 2. If BMCC equals one, then the estimated latents identify true latents up to permutation and block-diagonal transforms. Further details are in the Appendix (Section A.3).

**Non-overlapping perturbations** We first conducted experiments with one-sparse perturbations, the set $\Delta$ consists of $m = d$ one-sparse perturbations that span a $d$ dimensional space. In the context of the image experiments, these perturbations correspond to moving each ball individually along a single axis. The learner solves the identity in equation (3) using a set of random one-sparse perturbations $\Delta'$ that span a $d$ dimensional space. In Table 1, we used the low-dimensional synthetic data generating process to compare the effect of (i) applying all $m = d$ perturbations to each instance $z$ (following the DGP in (24)), against a more practical setting (ii) where a perturbation is selected uniform at random from $\Delta$ and applied to each instance $z$. The results for (i) are shown in black and the results for (ii) are shown in gray font in the *C-wise* (componentwise) column in Table 1. We observed high MCCs in both settings. The results were similar in the more challenging image-based experiments (see Table 3, C-wise column) with MCC scores $> 0.97$ for all the settings that we tested, as expected given the results presented in Theorem 1.

In our next experiments, the set of perturbations $\Delta$ comprised of $d$ 2-sparse non-overlapping perturbations that span a $d$ dimensional space. We repeated the same synthetic experiments as above with one and $d$ perturbations per instance. Under these assumptions we should expect to see that pairs of latents are separated blockwise but linearly entangled within the blocks (c.f. Theorem 2). We found this to be the case. The high BMCC numbers in Table 1 displayed under *B-wise* (blockwise) column (except for $d = 20$ and one perturbation per sample) show disentanglement between the blocks of latents. In Figure 2, the first two rows and columns show how the predicted latents corresponding to a block are correlated with their true counterpart (see Predicted $\tilde{Z}_i$ vs True $Z_i$) and the other latent in the block (Predicted $Z_1$ vs True $Z_2$ and vice versa). The plots in the last column show that the predicted latents did not bear a correlation with a randomly selected latent from outside the block.

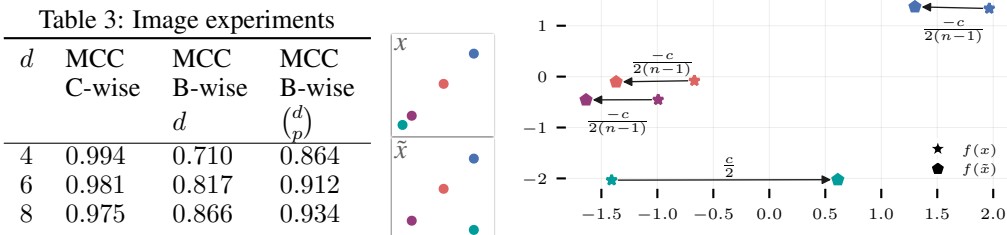

| Table 3: Image experiments | | | |
|---|---|---|---|
| $d$ | MCC C-wise | MCC B-wise $d$ | MCC B-wise $\binom{d}{p}$ |
| 4 | 0.994 | 0.710 | 0.864 |
| 6 | 0.981 | 0.817 | 0.912 |
| 8 | 0.975 | 0.866 | 0.934 |

Figure 3: *(Left)* Results for the image-base experiments. *(Centre)* Example images in which the bottom left ball is shifted to the right. *(Right)* A trained encoder's predictions for the two images shown in *(centre)*. The green ball prediction shifts right by $\approx \frac{c}{2}$ and the other balls left by $\approx \frac{c}{2(n-1)}$. For further illustrations, refer to the animations in the supplement.

**Overlapping perturbations** In this section, we experimented with blocks of size two that overlap in order to conform with the setting described in Theorem 4. We used the same distributions as before and only changed the type of perturbations. The low-dimensional synthetic results are summarized in Table 2. The results were largely as expected, with a strong correspondence between the predicted and true latents reflected by high MCC values.

On the image datasets (see Table 3), we found that the MCC scores depended on both the number of balls and how the blocks were selected. We compared two strategies for selecting blocks of latents to perturb: either select uniformly from all adjacent pairs $\mathcal{I} = \{(i \bmod d, i + 1 \bmod d)\}$ ($d$ blocks), or uniformly from all combinations of latent indices, $\mathcal{I} = \{(i, j) : i \in \{1, \ldots, d\}, j > i\}$ ($\binom{d}{2}$ blocks). The latter lead to higher MCC scores (ranging from 0.86 to 0.93) as it placed more constraints on the solution space. The dependence on the number of balls is more surprising. To investigate the implied entanglement from the lower MCC scores, we evaluated trained encoders on images where we kept $n_{\text{balls}} - 1$ balls in a fixed location and moved one of the balls (see Section A.3 in the Appendix for example images). If the coordinates were perfectly disentangled, the encoder should predict no movement for static balls. When the moving ball shifted by $c$ units, the predicted location of the static balls shifted by $\approx \frac{-c}{2(n_{\text{balls}}-1)}$ and the moving ball shifted $\approx \frac{c}{2}$ units. We further verified this claim and ran blockwise experiments with $n_{\text{balls}} = 10$ balls ($d = 20$) and got MCC scores of 0.930 and 0.969 for $d$ and $\binom{d}{2}$ blocks respectively. In the Appendix (Section A.3), we show that this solution is a stationary point, and we achieve a perfect MCC of one when $n_{\text{balls}} = \infty$. Finally, the code to reproduce the experiments presented above can be found at `https://github.com/ahujak/WSRL`.

## 5 Discussion and limitations

Our work presents the first systematic analysis of the role of sparsity in achieving latent identification under unknown arbitrary latent distributions. We assume that every sample (or at least every neighborhood of a sample) experiences the same set of perturbations. A natural question is how to extend our results to settings where this assumption may not hold. Data augmentation provides a rich source of perturbations; our results cover translations, but extending them to other forms of augmentation is an important future direction. We followed the literature on non-linear ICA (Hyvarinen et al., 2019) and made two assumptions – i) the map $g$ that mixes latents is injective, and ii) the dimension of the latent $d$ is known. We believe future works should aim to relax these assumptions. In reinforcement learning (RL) environments, the effects of actions can often be sparse. Therefore, we believe illustrating the efficacy of the proposed approach in RL environments (Ahmed et al., 2020) is an important direction to further the case of the proposed theory and methods in real-world applications.

## 6 Acknowledgements

We thank Sébastien Lachapelle and Anirudh Goyal for insightful discussions. Kartik Ahuja acknowledges the support from the IVADO postdoctoral fellowship. Jason Hartford acknowledges support from the Natural Sciences and Engineering Research Council of Canada (NSERC) and Recursion Pharmaceuticals. Yoshua Bengio acknowledges the support from CIFAR, Samsung and IBM.

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
