# A Appendix

We organize the Appendix into three sections. In Section A.1, we provide the proofs to all the propositions and the theorems. In Section A.2, we discuss how some of the proposed results can be extended. In Section A.3, we provide supplementary materials for the experiments.

## A.1 Proofs

We restate all the propositions and the theorems below for convenience. In the proofs that follow, we use $\Delta$ ($\Delta^{'}$) to denote the set of perturbations and the matrix of perturbations interchangeably (their usage is clear from the context). We start with the proof of Proposition 1, which follows the proof technique from Ahuja et al. (2022).

**Proposition 2.** *(Restatement of Proposition 1) If Assumptions 1, 2, and 3 hold, then the encoder that solves equation (3) (with $\Delta^{'}$ s.t. $\dim\left(\mathsf{span}(\Delta^{'})\right) = d$) identifies true latents up to an invertible affine transform, i.e. $\hat{z} = Az + c$, where $A \in \mathbb{R}^{d \times d}$ is an invertible matrix and $c \in \mathbb{R}^d$ is an offset.*

*Proof.* We simplify the identity in equation (3) as follows.

$$
\begin{aligned}
f(x) + \delta_i^{'} &= f(\tilde{x}_k) \\
f \circ g(z) + \delta_i^{'} &= f \circ g(\tilde{z}_k) \\
a(z) + \delta_i^{'} &= a(\tilde{z}) \\
a(z) + \delta_{i'}^{'} &= a(z + \delta_i)
\end{aligned}
\tag{6}
$$

In the above simplification, we use the following observation. Since $x$ and $\tilde{x}_k$ are generated from $g$ and $g$ is injective, we can substitute $x = g(z)$ and $\tilde{x}_k = g(\tilde{z}_k)$, where $\tilde{z}_k = z + \delta_k$.

For simplicity denote the last line in above equation (6) as

$$
a(z) + b^{'} = a(z + b). \tag{7}
$$

We take gradient of the function in the LHS and RHS of the above equation (7) separately w.r.t $z$. Consider the $j^{th}$ component of $a(z + b)$ denoted as $a_j(z + b)$. We first take the gradient of $a_j(z + b)$ w.r.t $z$

$$
\nabla_z a_j(z + b) = \left(\frac{dy}{dz}\right)^{\mathsf{T}} \nabla_y a_j(y), \tag{8}
$$

where $y = z + b$, $\nabla_y a_j(y)$ is the gradient of $a_j$ w.r.t $y$ and $\frac{dy}{dz}$ denotes the Jacobian of $y$ w.r.t $z$. We simplify the above further to get

$$
\nabla_z a_j(z + b) = \nabla_y a_j(y) = \nabla_y a_j(z + b). \tag{9}
$$

We can write the above for each component of $a$ as follows.

$$
\begin{aligned}
\left[\nabla_z a_1(z + b), \cdots, \nabla_z a_d(z + b)\right] &= \left[\nabla_y a_1(z + b), \cdots, \nabla_y a_d(z + b)\right] \\
&= \left[\nabla_y a_1(z + b), \cdots, \nabla_y a_d(z + b)\right] = J^{\mathsf{T}}(z + b),
\end{aligned}
\tag{10}
$$

where $J(z + b)$ is the Jacobian of $a$ computed at $z + b$. We equate the gradient of LHS and RHS in (7) to obtain

$$
a(z + b) = a(z) + b^{'} \implies J^{\mathsf{T}}(z + b) - J^{\mathsf{T}}(z) = 0. \tag{11}
$$

Consider row $j$ of this identity. For each $z \in \mathbb{R}^d$

$$\nabla a_j(z+b) - \nabla a_j(z) = 0 \implies \begin{bmatrix} \nabla_1^2 a_j(\theta_1) \\ \nabla_2^2 a_j(\theta_2) \\ \vdots \\ \nabla_d^2 a_j(\theta_d) \end{bmatrix} (b) = 0 \tag{12}$$

where $\nabla^2 a_j$ is the Hessian of $a_j$ and $\nabla_k^2 a_j(\theta_k)$ corresponds to the $k^{th}$ row of the Hessian matrix. Note that in the above expansion there is a different $\theta_k$ for each row (mean value theorem applied to each component of $\nabla a_j$ yields a different point $\theta_k$ on the line joinining $\tilde{z}$ and $\tilde{z} + b$). From Assumption 3 it follows that $\nabla_k^2 a_j(\theta_k)(b) = 0$ over a set with non-zero measure. Since $a_j$ is analytic $\nabla_k^2 a_j(z)(b)$ is also analytic (each component of the vector is a weighted sum of analytic functions). Therefore, we can conclude that $\nabla_k^2 a_j(z)(b) = 0$ for all $z$ (follows from Mityagin (2015)). We can make the same argument for each component $k$ and conclude that $\nabla^2 a_j(z)(b) = 0$. From the identity in equation (3), it follows that $\nabla^2 a_j(z)(\delta_j) = 0$ for all $j \in \{1, \cdots, d\}$ and since the set $\Delta = \{\delta_1, \cdots, \delta_d\}$ is linearly independent $\nabla^2 a_j(z) = 0$ for all $z$. This implies $a(z) = Az + c$.

We substitute this in equation (6) to get $A\Delta = \Delta'$, where $\Delta$ is the matrix of true perturbations and $\Delta'$ is the matrix of guessed perturbations (recall we stated above that we use $\Delta, \Delta'$ as sets and matrices interchangeably). We now need to show that $A$ is invertible. Suppose $A$ was not invertible, which implies the rank of $A \le n-1$. Following Assumption 2, rank of $\Delta$ is $n$. Note that rank of $\Delta'$ is also $n$. Note that if $E = FG$, where $E, F, G$ are three matrices, then $\mathsf{rank}(E) \le \min\{\mathsf{rank}(F), \mathsf{rank}(G)\}$. Following this identity, $\mathsf{rank}(\Delta') \le n-1$, which is a contradiction. Therefore, $A$ has to be invertible. This completes the proof.

$\square$

**Theorem 5.** *(Restatement of Theorem 1) If Assumptions 1-4 hold and the number of perturbations per example equals the latent dimension, $m = d$, then the encoder that solves equation (3) (with $\Delta'$ as one-sparse and $\dim\left(\mathsf{span}(\Delta')\right) = d$) identifies true latents up to permutation and scaling, i.e. $\hat{z} = \Pi\Lambda z + c$, where $\Lambda \in \mathbb{R}^{d \times d}$ is an invertible diagonal matrix, $\Pi \in \mathbb{R}^{d \times d}$ is a permutation matrix and $c$ is an offset.*

*Proof.* Since Assumptions 1, 2, and 3 hold, we can use Proposition 1 to obtain that any solution to equation (3) achieves affine identification guarantees w.r.t the true latents, i.e. $\hat{z} = Az + c$, where $\hat{z} = f(x)$, $z$ is the inverse image of $x$ ($x = g(z)$), $A \in \mathbb{R}^{d \times d}$ is an inverible matrix and $c \in \mathbb{R}^d$ is the offset vector.

Define $e_i = [0, \cdots, 1_i, \cdots 0]$ as the vector, which takes a value 1 at $i^{th}$ component and 0 everywhere else. Without loss of generality set of true perturbations is $\Delta = \{b_1 e_1, \cdots, b_d e_d\}$. Note that all $b_i$'s are non-zero as the span of $\Delta$ has a dimension $d$.

Denote the corresponding set of guesses from the agent are $\Delta' = \{c_1 e_{\pi(1)}, \cdots, c_d e_{\pi(d)}\}$, where $\pi : \{1, \cdots, d\} \to \{1, \cdots, d\}$ is a map used by the agent to guess the coordinate impacted by the perturbation. Note that since $\Delta'$ spans $d$ dimensions $\pi$ has to be a bijection $c_j$'s are non-zero as the span of $\Delta'$.

Take $b_j e_j \in \Delta$ and the corresponding guess $c_k e_k$ and substitute it in the relation $\hat{z} = Az + c$ to get

$$\begin{aligned} \hat{z} &= Az + c, \\ \hat{z} + c_k e_k &= A(z + b_j e_j) + c, \\ c_k e_k &= b_j A e_j, \\ \frac{c_k}{b_j} e_k &= A e_j. \end{aligned} \tag{13}$$

Since $\pi$ is a bijection, for every $j$ there is a unique $k$ in the RHS above. From the above equation, we gather that the $j^{th}$ column of $A$ is $\frac{c_k}{b_j} e_k$. We apply this to all the columns and conclude that $\hat{z} = \Pi\Lambda z + c$, where $\Lambda$ is a diagonal matrix and $\Pi$ is a permutation matrix decided based on the bijection $\pi$ (($\Pi_k = e_{\pi(k)}$), where $\Pi_k$ is the $k^{th}$ colum of the matrix). $\square$

**Theorem 6.** *(Restatement of Theorem 2) If Assumptions 1-3, 5, 6 hold, then the encoder that solves equation (3) (where $\Delta^{'}$ is p-sparse, $\dim\left(\mathrm{span}\left(\Delta^{'}\right)\right) = d$) identifies true latents up to permutation and block-diagonal transforms, i.e. $f(x) = \hat{z} = \Pi\tilde{\Lambda}z + c$, where $\tilde{\Lambda} \in \mathbb{R}^{d \times d}$ is an invertible block-diagonal matrix with blocks of size $p \times p$, $\Pi \in \mathbb{R}^{d \times d}$ is a permutation matrix and $c \in \mathbb{R}^d$ is an offset.*

*Proof.* Since Assumptions 1, 2, and 3 hold, we can use Proposition 1 to obtain that any solution to equation (3) achieves affine identification guarantees w.r.t the true latents, i.e. $\hat{z} = Az + c$, where $\hat{z} = f(x)$, $z$ is the inverse image of $x$ ($x = g(z)$), $A \in \mathbb{R}^{d \times d}$ is an inverible matrix and $c \in \mathbb{R}^d$ is the offset vector.

We start the proof by assuming that the agent knows the blocks that are impacted under each perturbation, i.e., for each $i \in \mathcal{I}$, the agent knows the block of the latents that are impacted denoted as $\mathcal{A}_i$. We relax this assumption later.

Following Assumption 5, we know that perturbations are $p$-sparse, blockwise and non-overlapping. Without loss of generality, we can assume that the different groups on which perturbations in $\Delta$ act are given as $\{1, \cdots, p\}$, $\{p+1, \cdots, 2p\}$ and so on. Consider a perturbation $\delta_i$, which belongs to Group 1 and impacts the latents $\{1, \cdots, p\}$. For this perturbation, the agent selects $\delta_i^{'}$, which shares the same sparsity pattern. Therefore, the first $p$ elements of $\delta_i^{'}$ and $\delta_i$ are both non-zero and the rest of the elements are zero. Under these assumptions, we can write the relationship between true and guessed perturbations as follows.

$$\hat{z} + \delta_i^{'} = A(z + \delta_i) + c$$
$$\delta_i^{'} = A\delta_i \tag{14}$$

Denote the first $p$ elements of row $k$ of matrix $A$ as $a_k[1:p]$ and the first $p$ elements of the vector $\delta_i$ as $\delta_i[1:p]$. For $k > p$, we use the equation (14) to get $a_k[1:p]^\mathsf{T}\delta_i[1:p] = 0$.

For all perturbations in Group 1, we can write the same condition, i.e., $a_k[1:p]^\mathsf{T}\delta_i[1:p] = 0$. Since the perturbations in Group 1 span a $p$ dimensional space (following Assumption 2, 5), we get that $a_k[1:p] = 0$. Therefore, $a_k[1:p] = 0$ for all $k > p$.

Let $q$ denote the number of perturbations in Group 1, where $q \geq p$. For all $k \leq p$ we can solve for the first $p \times p$ block using the perturbations guessed by the agent and the true perturbations in Group 1. Denote the first $p \times p$ block of $A$ as $A[1:p, 1:p]$ and the first $p$ components of the $q$ perturbations in Group 1 as $\Delta[1:p, 1:q]$. Similarly, the first $p$ components of the $q$ perturbations guessed by the learner is denoted as $\Delta^{'}[1:p, 1:q]$. We now need to show that the block $A[1:p, 1:p]$ is invertible. From the above equation in (14), we get

$$A[1:p, 1:p]\Delta^{'}[1:p, 1:q] = \Delta[1:p, 1:q]$$

.

where $q$ is the number of perturbations in Group 1.

Since rank of $\Delta[1:p, 1:q]$ and $\Delta^{'}[1:p, 1:q]$ is $p$, the rank of $A[1:p, 1:p]$ cannot be less than $p$ or else it would lead to a contradiction. This shows that $A[1:p, 1:p]$ is invertible. We derived the properties of the first $p$ columns of matrix $A$. For Group 2, we similarly obtain that $A[p+1:2p, p+1:2p]$ is an invertible matrix and rest of the elements in columns $\{p+1, \cdots, 2p\}$ are zero. Due to symmetry of the setting, we can apply the same argument to all the other blocks as well. Therefore, we conclude that $A$ is block-diagonal and invertible. This leads to the conclusion that $\hat{z} = \tilde{\Lambda}z + c$, where $\tilde{\Lambda} \in \mathbb{R}^{d \times d}$ and $c \in \mathbb{R}^d$.

So far we assumed that the agent knows how the interventions in $\{1, \cdots, m\}$ impact the blocks $\{\mathcal{A}_1, \cdots, \mathcal{A}_m\}$. Under Assumption 6, the agent knows the groups of the perturbations only. For perturbations $\{\delta_1, \cdots, \delta_p\}$ in Group 1 that impact $\{1, \cdots, p\}$, the agent guesses $\{\delta_1^{'}, \cdots, \delta_p^{'}\}$. Note that perturbations in $\{\delta_1^{'}, \cdots, \delta_p^{'}\}$ impact the same block of length $p$ with indices given as $\{\alpha_1, \cdots, \alpha_p\}$. Recall the first $p$ elements of row $k$ of matrix $A$ and vector $\delta_i$ are denoted as $a_k[1:p]$ and $\delta_i[1:p]$ respectively. There exist $d-p$ rows in $A$ for which we get $a_k[1:p]^\mathsf{T}\delta_i[1:p] = 0$. Thus $a_k[1:p] = 0$ for

all these rows. The first $p$ elements of remaining $p$ form a square matrix denoted as $A[\alpha_1 : \alpha_p, 1 : p]$, where $\{\alpha_1, \cdots, \alpha_p\}$ are the indices guessed by the agent for the block corresponding to Group 1. $A[\alpha_1 : \alpha_p, 1 : p]$ satisfies

$$A[\alpha_1 : \alpha_p, 1 : p]\Delta[1 : p, 1 : q] = \Delta'[\alpha_1 : \alpha_p, 1 : q]$$

where $\Delta'[\alpha_1 : \alpha_p, 1 : q]$ is the matrix of non-zero components of the $q$ perturbation vectors that the agent guesses. Using the same argument as above, we can argue that $A[\alpha_1 : \alpha_p, 1 : p]$ is invertible. We have derived the properties of first $p$ columns of $A$. We apply the same argument to other groups as well. Since the agent selects a set of unique $p$ indices for each group, we obtain that the matrix $A$ can be factorized as a permutation matrix times a block diagonal matrix. The first $p$ rows of the permutation matrix with index $\{1, \ldots, p\}$ have ones at locations $\{\alpha_1, \cdots, \alpha_p\}$ and so on. As a result, we get that $\hat{z} = \Pi\tilde{\Lambda}z + c$

This completes the proof. $\qquad\square$

**Theorem 7.** *(Restatement of Theorem 3) Suppose Assumptions 1, 3, 6 and 7 hold. Consider the subsets $\mathcal{I}_1$ and $\mathcal{I}_2$ that satisfy Assumption 7. For every pair of blocks, $\mathcal{B}^1 \in \mathcal{B}_{\mathcal{I}_1}$ and $\mathcal{B}^2 \in \mathcal{B}_{\mathcal{I}_2}$, the encoder that solves equation (3) (where $\Delta'$ is $p$-sparse, $\mathsf{dim}\Big(\mathsf{span}\big(\Delta'\big)\Big) = d$) identifies latents in each of the blocks $\mathcal{B}^1 \cap \mathcal{B}^2$, $\mathcal{B}^1 \setminus \mathcal{B}^2$, $\mathcal{B}^2 \setminus \mathcal{B}^1$ up to invertible affine transforms.*

*Proof.* Following Assumption 7, we know that there exists at least two subsets $\mathcal{I}_1$ and $\mathcal{I}_2$ that satisfy blockwise non-overlapping perturbations. Like in the previous proof, we start this proof also with the case where the agent knows the exact sparsity pattern in the perturbations. We relax this assumption in a bit. Consider a block $\mathcal{B}^1 = \{\beta_1, \cdots, \beta_p\}$ impacted by the perturbations in $\mathcal{I}_1$. Since $\mathcal{I}_1$ is blockwise and non-overlapping, we can follow the analysis in the first part of the previous theorem to get $[\hat{z}_{\beta_1}, \cdots, \hat{z}_{\beta_p}]$ is an invertible affine transform of $[z_{\beta_1}, \cdots, z_{\beta_p}]$. Hence, the latents in each of the blocks $\mathcal{B}^1 \in \mathcal{G}_{\mathcal{I}_1}$ are identified up to an afffine transform. Similarly, each block $\mathcal{B}^2 \in \mathcal{G}_{\mathcal{I}_2}$ is identified up to an affine transform. Consider an element $i \in \mathcal{B}^1 \cap \mathcal{B}^2$. $\hat{z}_i$ can be expressed as an affine transform of two different blocks of latents $z^1$ and $z^2$. $z^1$ and $z^2$ share some components, we denote them as $z^{12}$. The components exclusive to $z^1$ ($z^2$) is denoted as $z^{11}$ ($z^{22}$).

We write this condition as follows.

$$\begin{aligned}
\hat{z}_i &= a_1^\mathsf{T} z^{11} + a_2 z^{12} + a_3 \\
\hat{z}_i &= b_1^\mathsf{T} z^{22} + b_2 z^{12} + b_3 \\
a_1^\mathsf{T} z^{11} &+ (a_2 - b_2)^\mathsf{T} z^{12} - b_1^\mathsf{T} z^{22} = b_3 - a_3
\end{aligned} \qquad (15)$$

If $[a_1, a_2 - b_2, b_1]$ is non-zero, i.e., at least one element is non-zero, then the range of LHS is $\mathbb{R}$. But the range of the RHS is a constant. Therefore, for the above to be true $[a_1, a_2 - b_2, b_1] = 0$ and that implies $a_3 = b_3$. As a result, the linear entanglement is now confined to only the intersecting variables $z^{12}$. We can repeat this argument for all elements in $\mathcal{B}^1 \cap \mathcal{B}^2$.

In the proof so far, we relied on the assumption that the components impacted by each intervention $i \in \mathcal{I}$ are known. We now relax this assumption and work with assumption that was used in the previous theorem (Assumption 6), which states that the agent knows the group label of each perturbation.

Consider the latents in the block $\mathcal{B}^1 \in \mathcal{G}_{\mathcal{I}_1}$, which we denote as $z^1$. We apply Theorem 2 to this block. Let the set of estimated latents that affine identify $\mathcal{B}^1$ be $\hat{z}^1 = [\hat{z}_{\alpha_1}, \cdots, \hat{z}_{\alpha_p}]$, where $\{\alpha_1, \cdots, \alpha_p\}$ is the set of indices in $\hat{z}$. We write this as $[\hat{z}_{\alpha_1}, \cdots, \hat{z}_{\alpha_p}] = A^1 z^1 + c^1$. $\tilde{\mathcal{B}}^1$ denotes the set of remaining latents not in the block $\mathcal{B}^1$. We denote the latents in the block $\tilde{\mathcal{B}}^1$ as $z_c^1$. Following Theorem 2, we get that the remaining elements of $\hat{z}$ other than $\hat{z}^1$, which we denote as $\hat{z}_c^1$, affine identify the latents $z_c^1$ in the block $\tilde{\mathcal{B}}^1$.

Similarly, consider the latents in the group $\mathcal{B}^2 \in \mathcal{G}_{\mathcal{I}_2}$ denoted as $z^2$. $\hat{z}^2 = [\hat{z}_{\beta_1}, \cdots, \hat{z}_{\beta_p}]$ denotes the latents that affine identify $z^2$. $\tilde{\mathcal{B}}^2$ is the set of remaining latents. The remaining elements of $\hat{z}$ other than $\hat{z}^2$ are denoted as $\hat{z}_c^2$. $\hat{z}_c^2$ affine identifies the latents in the block $\tilde{\mathcal{B}}^2$, which are denoted as $z_c^2$.

The latents $z^{11} \in \mathcal{B}^1 \setminus \mathcal{B}^2$, $z^{12} \in \mathcal{B}^1 \cap \mathcal{B}^2$, and $z^{22} \in \mathcal{B}^2 \setminus \mathcal{B}^1$. Consider a latent that is shared between $\hat{z}^1$ and $\hat{z}^2$. Using the same analysis from equation (15), we show that such an element puts a non-zero weight only on $z^{12}$. Therefore, all the latents shared between $\hat{z}^1$ and $\hat{z}^2$ have a non-zero weight on $z^{12}$. Now consider a component of $\hat{z}^1$ denoted as $\hat{z}_{\alpha_k}$, which is not present in $\hat{z}^2$. We can write the affine identification condition as

$$\hat{z}_{\alpha_k} = c_1^\mathsf{T} z^{11} + c_2^\mathsf{T} z^{12} + c_3 \tag{16}$$

We selected $\hat{z}_{\alpha_k}$, which is not present in $\hat{z}^2$. Since $\hat{z}_{\alpha_k}$ is in $\hat{z}_c^2$, we have

$$\hat{z}_{\alpha_k} = d_1^\mathsf{T} z_c^2 + d_3 \tag{17}$$

If we take a difference of the above two equations (16) and (17), we get that $c_2$ is equal to zero (see the justification below).

$$d_1^\mathsf{T} z_c^2 + d_3 - c_1^\mathsf{T} z^{11} - c_2^\mathsf{T} z^{12} - c_3 = 0 \tag{18}$$

Note that there is no term associated with $z^{12}$ in equation (17) as $z_c^2$ is the set of elements not in $z^2$. Now since the above equation (17) holds for all $z$, we get $c_2 = 0$.

From the above analysis we conclude that the latents in $\hat{z}^1$ can be divided into two parts i) the latents that are shared with $\hat{z}^2$; these latents are an affine transform of $z^{12}$, ii) the latents that are not shared with $\hat{z}^2$; these latents are an affine transform of $z^{11}$. We write this condition as

$$\hat{z}^1 = \begin{bmatrix} e_1 & 0 \\ 0 & e_2 \end{bmatrix} \begin{bmatrix} z^{11} \\ z^{12} \end{bmatrix} + e_3 \tag{19}$$

Similarly, we get

$$\hat{z}^2 = \begin{bmatrix} f_1 & 0 \\ 0 & f_2 \end{bmatrix} \begin{bmatrix} z^{22} \\ z^{12} \end{bmatrix} + f_3 \tag{20}$$

We have already discussed above that $f_2 = e_2$ and the latter half of $f_3$ corresponding to $z^{12}$ is equals corresponding half of $e_3$.

From the previous theorem, we know that the matrices in the above equations (19) and (20) are invertible. Thus if $z^{12}$ has $q$ components, then $e_2$ is an invertible $q \times q$ matrix and $e_1$ is an invertible $p - q \times p - q$ matrix. This establishes the affine identification of the smaller blocks obtained by intersection of the blocks across two sets of non-overlapping blockwise perturbations. This completes the proof. □

**Theorem 8.** *(Restatement of Theorem 4) Suppose Assumptions 1, 3, 6 and 8 hold, then the encoder that solves the identity in equation (3) (where $\Delta^{'}$ is $p$-sparse, $\mathsf{dim}\left(\mathsf{span}\left(\Delta^{'}\right)\right) = d$) identifies true latents up to permutations and scaling, i.e., $\hat{z} = \Pi\Lambda z + c$, where $\Pi \in \mathbb{R}^{d \times d}$ matrix and $\Lambda \in \mathbb{R}^{d \times d}$ is a diagonal matrix.*

*Proof.* In the above theorem, we use a set of perturbations $\mathcal{I}$ that are $p$-sparse and satisfy the following property. The first $d - (p - 1)$ blocks are $\{i, \cdots, i + p - 1\}$ from $i = 1$ to $i = d - p + 1$. The remaining $p - 1$ blocks are $\{i, \cdots, (i + p - 1) \bmod (d + 1) + 1\}$ from $i = d - p + 2$ to $d$. In the $d$ blocks each latent component $i$ is the first element of the block exactly once and also the last component exactly once.

Construct a partition of perturbations $\mathcal{I}_1$ with contiguous blocks $\{k, \cdots, k + p - 1\}$ and so on. Similarly, construct a partition of perturbations $\mathcal{I}_2$ $\{k - (p - 1), \cdots, k\}$ and so on. Note that $k$ is the first element of its block in $\mathcal{I}_1$ and it is the last element of its block in $\mathcal{I}_2$. We can apply the Theorem 3 to conclude that $k^{th}$ component is identified up to scaling and permutation error. We can state the same for all the components. This completes the proof. □

### A.2 Extensions

#### A.2.1 Extending Theorem 1

In Theorem 1, we assumed that the number of perturbations $m$ is equal to the number of latent dimensions $d$. Suppose the number of perturbations is larger than $d$. We subsample $d$ distinct perturbation indices from $\{1, \cdots, m\}$. We solve the identity with the data generated under sub-sampled perturbations in equation (3) with one-sparse guesses. If a solution exists, then we can continue to use the analysis in Theorem 1. If a solution does not exist, we sub-sample again and solve the identity in equation (3) until we find a solution.

In Theorem 1, we assumed that the learner knows that the perturbations in equation (24) are one sparse. Suppose the learner instead guesses that the perturbations are $p$-sparse, where $1 < p < d$ and $d \bmod p = 0$. In this case, we can use analysis similar to Theorem 2 and guarantee blockwise identification, where the blocks are of size $p \times p$.

#### A.2.2 Extending Theorem 2

In this section, we discuss how we can relax Assumption 6. We first show how to extend Theorem 2 to this setting. Later we describe how the same ideas can be extended to setting presented in Theorem 4. In this section, we propose a sparsity test, which would be used to test if the encoder learned is $p$-sparse or not. For the rest of the section, we assume that $d \bmod p = 0$. The number of blocks is $r = \frac{d}{p}$.

We take each sample point $(x, \tilde{x}_1, \cdots, \tilde{x}_m)$ and divide it into two parts. We keep the first $d$ perturbations in one set $(\tilde{x}_1, \cdots, \tilde{x}_d)$ to train the encoder and we use the remaining $(\tilde{x}_{d+1}, \cdots, \tilde{x}_m)$ for checking sparsity. We refer to the first $d$ perturbations as training perturbations and the remaining perturbations as validation perturbations.

**Assumption 9.** $\{\delta_i\}_{i=1}^d$ *is the set of training perturbations, which are p-sparse, blockwise and non-overlapping.* $\{\delta_i\}_{i=d+1}^m$ *is the set of validation perturbations, which are p-sparse, blockwise and non-overlapping. The training perturbations span* $\mathbb{R}^d$.

Consider $d$ perturbations and represent them as follows $\Delta_d$

$$
\Delta_d = \begin{bmatrix} \Delta_{11} & \Delta_{12} & \cdots, \Delta_{1r} \\ \Delta_{12} & \Delta_{22} & \cdots, \Delta_{2r} \\ & \vdots & \\ \Delta_{r1} & \Delta_{22} & \cdots, \Delta_{rr} \end{bmatrix} \tag{21}
$$

where $\Delta_{ij}$ is $p \times p$ matrix. Without loss of generality under Assumption 9, we can write $\Delta_d$ as a blockdiagonal matrix such that all matrices $\Delta_{ij} = 0$ for all $i \neq j$.

We write the inverse of $\Delta_d$ as

$$
\Delta_d^{-1} = \begin{bmatrix} \tilde{\Delta}_{11} & \tilde{\Delta}_{12} & \cdots, \tilde{\Delta}_{1r} \\ \tilde{\Delta}_{12} & \tilde{\Delta}_{22} & \cdots, \tilde{\Delta}_{2r} \\ & \vdots & \\ \tilde{\Delta}_{r1} & \tilde{\Delta}_{22} & \cdots, \tilde{\Delta}_{rr} \end{bmatrix} \tag{22}
$$

**Assumption 10.** *Each element in the matrix along the diagonal of* $\Delta_d^{-1}$ *is non-zero, i.e.,* $\forall k \in \{1, \cdots, \frac{d}{p}\}, \forall i, j \in \{1, \cdots, p\}, \tilde{\Delta}_{kk}[i, j] \neq 0$

**Assumption 11.** $\Delta'$ *consists of d perturbations, which are p-sparse, blockwise and non-overlapping.*

We write the $d$ corresponding perturbations that the agent guesses in the form of a matrix as

$$
\Delta_d' = \begin{bmatrix} \Delta_{11}' & \Delta_{12}' & \cdots, \Delta_{1r}' \\ \Delta_{12}' & \Delta_{22}' & \cdots, \Delta_{2r}' \\ & \vdots & \\ \Delta_{r1}' & \Delta_{22}' & \cdots, \Delta_{rr}' \end{bmatrix} \tag{23}
$$

where $\Delta'_{ij}$ is $p \times p$ matrix.

Define an indicator mask of the underlying matrix $\Delta'$; it takes a value one wherever there is a non-zero entry and zero otherwise. Define the set of all the masks for $\Delta'$ that satisfy the above assumption (Assumption 11) as $\mathcal{M} = \{1, \cdots, n_{\mathsf{masks}}\}$. Now under the Assumption 9, we get that the validation perturbations are blockwise and non-overlapping as well (though they are not required to span the blocks). We now formalize a simple iterative procedure in which the learner searches over masks that are compliant with the assumption above (Assumption 9)

In the sparsity test, we take a trained encoder and check if for each of the perturbations in the validation set, it ensures only $p$ components change. If for any perturbation more than $p$ estimated components change, then the encoder fails the test.

**Joint mask search and encoder learning**

- Select candidate mask $i$ from $\mathcal{M}$. Fill the non-zero entries with random values from some distribution $\mathbb{P}_M$ (we assume that $\mathbb{P}_M$ has no mass on zero) to generate a candidate $\Delta'$

- Solve the identity in equation (3) using samples from the perturbations selected in the step above $\Delta'$. Check for $p$-sparsity on the set of validation perturbations. If the solution is at most $p$-sparse on all the validation perturbations, then select the encoder. If the solution fails, then $i = i + 1$ and go to step one.

The mask search procedure described above requires brute force search over many masks. Even though the procedure is computationally intractable it helps demonstrate that knowledge of sparsity can suffice (See Theorem 9 below).

**Theorem 9.** *Suppose Assumptions 1, 3, 9, 10, and 11 hold, then an encoder that is output learned following the joint mask search and encoder learning procedure above identifies latents up to permutation and block-diagonal transforms with probability one.*

*Proof.* We take the encoder $f(x)$ learned from joint mask search and encoder learning procedure described above. Following Assumptions 1, 3, 9 and 11, we obtain that $f(x) = \hat{z} = Az + c$, where $x = g(z)$, $A$ is an invertible matrix and $c$ is an offset. Following the analysis in Proposition 1, we obtain $A$ matrix is given as $A = \Delta'_d \Delta_d^{-1}$ (substitue $\hat{z} = Az + c$, $\hat{z} + \Delta_d = Az + c + A\Delta'_d$). We index the matrix in terms of the blocks.

The matrix at location $(i, j)$ is $A_{ij} = \Delta'_{ij}\tilde{\Delta}_{jj}$ (since $\Delta$ is a blockdiagonal matrix, i.e., $\Delta_{ij} = 0$ for $i \neq j$ but $\Delta_{ii} \neq 0$). Each column of $\Delta'$ consists of $p$ non-zero entries. Using this and $A_{ij} = \Delta'_{ij}\tilde{\Delta}_{jj}$ we obtain that the number of non-zero entries in each column of $A$ are at least $p$. We write $A_{ij}[k, q] = \sum_l \Delta'_{ij}[k, l]\tilde{\Delta}_{jj}[l, q]$. Since $\Delta'_{ij}[k, l]$ and $\tilde{\Delta}_{jj}[l, q]$ both take non-zero value, the first term in the above summation is non-zero. Since the other terms depend on random variables drawn independently, the probability that the sum equals zero is zero. Therefore, for each of the $p$ indices $k$ where the mask is non-zero, the $A_{ij}[k, q]$ is non-zero.

Suppose at least one column block of $A$, say $jp + 1 : (j + 1)p$, contains two columns which exhibit a different sparsity pattern. Since there are at least two columns which share a different sparsity pattern, there is at least one row where only one of them is zero and other is non-zero. Therefore, in this column block we have at least $p + 1$ rows which have at least one non-zero element. The encoder passed the sparsity test, i.e., for all the perturbations on blocks of the form $jp + 1 : (j + 1)p$ we have at most $p$-sparse output. Therefore, at least one of the $p + 1$ rows has to multiply with the block and output a zero, which is a zero probability event (since the non-zero elements of $A$ matrix are all continuous random variables). Thus if any contiguous block has different sparsity pattern across columns, then the encoder is selected with probablity zero. Thus from this we can conclude that for a selected encoder, each column block exhibits a sparsity pattern that is same across all the columns in the block. To ensure that $A$ is an invertible, all blocks exhibit a non-overlapping sparsity pattern. Therefore, $A$ is permutation times a diagonal matrix. We now illustrate what choices of $\Delta'$ lead to an $A$ that passes the sparsity test. If for every $i$ there exists a unique $j$ for which $\Delta'_{ij}$ is invertible and every other value of $j$, $\Delta'_{ij} = 0$, then $A$ is permutation times a diagonal matrix. This completes the proof. $\qquad \square$

In this section, we showed that we we do not need to make Assumption 6 and the knowledge of sparsity suffices to do blockwise identification. Following similar analysis as above, we can extend Theorem 4 as well.

### A.2.3 Extension to stochastic perturbations

In the DGP considered in Assumption 1, we assumed that the perturbations are determinstic. We now consider stochastic perturbations.

**Assumption 12.** *The DGP follows*

$$z_i \sim \mathbb{P}_Z, n_{ik} \sim \mathbb{P}_{N_k} \; \forall k \in \mathcal{I}, \;\; x_i \leftarrow g(z_i) \qquad \tilde{z}_{ik} \leftarrow z_i + \delta_k + n_{ik}, \forall k \in \mathcal{I} \qquad \tilde{x}_{ik} \leftarrow g(\tilde{z}_{ik}), \forall k \in \mathcal{I}$$
(24)

*where $g$ is injective and analytic, and $Z$ is a continuous random vector with full support over $\mathbb{R}^d$, $\mathbb{P}_{N_k}$ is the noise distribution for the $k^{th}$ perturbation.*

**Assumption 13.** *The perturbations in $\Delta = \{\delta_1, \cdots, \delta_m\}$ are one-sparse. Further, the noise vectors are also one-sparse and follow the same sparsity pattern, i.e. $n_{ik}$ follows the same sparsity pattern as the perturbation vector $\delta_k$ to which they are added.*

The above two assumptions can be understood as follows. Each perturbation is one sparse, i.e., under each perturbation one component of the latent $z$ changes by a fixed amount plus some noise. In the data generation process described above $\tilde{x}_{ik}$ corresponds to the $k^{th}$ perturbation of instance $x_i$. We write $X$ for the random vector corresponding to unperturbed observation and $\tilde{X}_k$ as the random vector associated with the $k^{th}$ perturbation. Denote the distribution of $k^{th}$ perturbation conditional on $X$ as $\mathbb{P}(\tilde{X}_k|X)$. The learner guesses the perturbation $\hat{x}_{ik}$ for instance $x_i$ as follows

$$\hat{x}_{ik} = f^{-1}\Big(f(x_i) + \delta_k' + n_{ik}'\Big)$$
(25)

where $f$ is the encoder (assumed to be bijective here) used by the learner, $\delta_k'$ is the perturbation guessed by the learner and $n_{ik}'$ is the noise sampled by the learner. We write the random variable based version of the above relationship as follows.

$$\hat{X}_k = f^{-1}\Big(f(X) + \delta_k' + N_k'\Big)$$
(26)

The learner's goal is to satisfy the following identity

$$\mathbb{P}(\hat{X}_k|X) = \mathbb{P}(X_k|X), \; \forall k \in \mathcal{I}$$

$$f^{-1}\Big(f(X) + \delta_k' + N_k'\Big) \stackrel{d}{=} g(Z + \delta_k + N_k), \; \forall k \in \mathcal{I}$$

$$f(X) + \delta_k' + N_k' \stackrel{d}{=} f(X_k) = f \circ g(Z + \delta_k + N_k), \; \forall k \in \mathcal{I}$$

$$a(Z) + \delta_k' + N_k' \stackrel{d}{=} a(Z + \delta_k + N_k), \; \forall k \in \mathcal{I}$$
(27)

where $\stackrel{d}{=}$ denotes equality in distribution. The above identity is the same as the equivariance in distribution condition arrived at in Ahuja et al. (2022). We now see how sparsity in the changes can be exploited to guarantee strong identification similar to our result in Theorem 1.

$$a(Z + \delta_k + N_k) = a(Z) + \begin{bmatrix} J_1(Z_1') \\ J_2(Z_2') \\ \vdots \\ J_d(Z_d') \end{bmatrix} (\delta_k + N_k)$$
(28)

where $J$ corresponds to the Jacobian of $a$. In the above equation (28), we carried out the first order Taylor expansion. We further simplify the equivariance in distribution to get the following.

$$a(Z) + \begin{bmatrix} J_1(Z_1^{'}) \\ J_2(Z_2^{'}) \\ \vdots \\ J_d(Z_d^{'}) \end{bmatrix} (\delta_k + N_k) \stackrel{d}{=} a(Z) + \delta_k^{'} + N_k^{'}$$

$$\begin{bmatrix} J_1(Z_1^{'}) \\ J_2(Z_2^{'}) \\ \vdots \\ J_d(Z_d^{'}) \end{bmatrix} (\delta_k + N_k) \stackrel{d}{=} \delta_k^{'} + N_k^{'}$$

(29)

Suppose that the first component of $\delta_k + N_k$ is non-zero. Due to one-sparsity we know that all the remaining components are zero. Suppose $\delta_k^{'} + N_k^{'}$ also have the same sparsity pattern as $\delta_k + N_k$ (we can arrive at qualitatively the same result that follows even in the absence of this assumption). As a result, $[J_{21}(Z_2^{'}), \cdots, J_{d1}(Z_d^{'})] = 0$. Suppose $a$ is analytic. As a result, the Jacobian $J$ is analytic as well. Further, suppose that the support of $Z_j^{'}$ for all $j \geq 2$ has a non-zero measure (this condition is the extension of Assumption 3 from deterministic case to the stochastic case). From (Mityagin, 2015) it follows that $[J_{21}(z), \cdots, J_{d1}(z)] = 0$ is identically zero. Since the identity in equation (28) holds for all $k$ we can conclude that the Jacobian of $a$ is a diagonal matrix. Since $\hat{z} = a(z)$, we can conclude that changes to one component of $z$ impact exactly one component of $\hat{z}$ and not the rest. Thus we can conclude we have perfect disentanglement. So far we analyzed the case where perturbations are one-sparse. We can generalize the above argument to non-overlapping blockwise perturbations following similar arguments to the deterministic case as well.

### A.2.4 Extension to non-linear mechanisms for perturbations

In the main body of the paper, we assumed that the data is generated under sparse and fixed perturbations. We now extend our analysis to the case where different perturbation can be applied to different points $z$. A mechanism $m : \mathbb{R}^d \to \mathbb{R}^d$ takes as input the latent and outputs the perturbation vector. We call a mechanism $p$-sparse, if it only changes $p$ components out of the $d$ latents, i.e., $\forall z \in \mathbb{R}^d$, $\exists\, d - p$ components of $z$ which remain unchanged on application of $m$. We write this data generation process as follows.

**Assumption 14.** *The DGP follows*

$$z \sim \mathbb{P}_Z, \; \tilde{z}_k \leftarrow z + m_k(z) \; \forall k \in \mathcal{I}, \tilde{x}_k \leftarrow g(\tilde{z}_k) \; \forall k \in \mathcal{I},$$

(30)

*where $g$ is injective and analytic, and $Z$ is a continuous random vector with full support over $\mathbb{R}^d$, $m_k$ is the $k^{th}$ perturbation mechanism.*

**Assumption 15.** *Each $m_k$ is one-sparse. For each latent dimension $i \in \{1, \cdots, d\}$, $\exists$ a mechanism $m_k \in \{m_1, \cdots, m_m\}$ that changes that latent dimension.*

Recall that $a = f \circ g$, where $f$ is the encoder that the learner uses. We state the assumption on $a$ below.

**Assumption 16.** *$a$ is an analytic function. For each component $i \in \{1, \cdots, d\}$ of $a(z)$ and each component $j \in \{1, \cdots, d\}$ of $z$, define the set $\mathcal{S}^{ij} = \{\theta \mid a_i(z + b) = a_i(z) + \nabla_j a_i(\theta)b, z \in \mathbb{R}^d\}$, where $b$ is a fixed vector in $\mathbb{R}^d$. Each set $\mathcal{S}^{ij}$ has a non-zero Lebesgue measure in $\mathbb{R}^d$.*

For each perturbation, the learner uses a $m_k^{'} : \mathbb{R}^n \to \mathbb{R}^d$ to guess the changes caused by the true mechanism $m_k$. We write the identity that the learner solves as follows. $\forall k \in \{1, \cdots, m\}$ and $\forall (x, \tilde{x}_k)$ generated by the DGP in Assumption 14

$$f(\tilde{x}_k) = f(x) + m_k^{'}(x).$$

(31)

**Theorem 10.** *If Assumption 14, 15, and 16, hold, then the solution to equation (31) (with one-sparse $m_k^{'}$), satisfies $\hat{z} = \Pi\Lambda(z) + c$, where $\Pi$ is a permutation matrix, $\Lambda(z) = \mathsf{diag}[\lambda(z_1), \cdots, \lambda(z_d)]$ is a function whose each component exactly depends on one latent dimension.*

*Proof.*

$$f(\tilde{x}_k) = f(x) + m'_k(x),$$
$$a(\tilde{z}_k) = a(z) + m^\dagger_k(z),$$
$$a(z + m_k(z)) = a(z) + m^\dagger_k(z),$$

(32)

where $m^\dagger_k = m'_k \circ g$. We drop $k$ from the above equation for ease of presentation and get

$$a(z + m(z)) = a(z) + m^\dagger(z)$$

(33)

We do a Taylor expansion of $a$ around $z$ to get

$$a(z) + [J_1(z'_1), J_2(z'_2), \cdots, J_d(z'_d)]m(z) = a(z) + m'(z)$$
$$\implies [J_1(z'_1), J_2(z'_2), \cdots, J_d(z'_d)]m(z) = m'(z)$$

(34)

Suppose $m(\cdot)$ and $m'(\cdot)$ are both one-sparse and impact the first component of $z$. From the above it follows that all elements of $[J_{21}(z'_1), \cdots, J_{d1}(z'_d)]$ are zero except $J_{11}(z'_j)$. Since the above holds true for all $z$, $[J_{21}(z'_1), \cdots, J_{d1}(z'_d)]$ would be zero on a set of measure non-zero. Thus $[J_{21}, \cdots, J_{d1}]$ is identically zero. We can repeat the same argument for all the columns and conclude that in each column all rows except one are zero. From this we can conclude that $a(z) = \Pi\Lambda(z) + c$. $\qquad\square$

We can follow the same line of reasoning and argue for blockwise identification as well.

### A.2.5  Connection with causal interventions

In the DGP in equation (24), we assumed that $Z$ is sampled from any distribution $\mathbb{P}_Z$. We now consider a special case, where $Z = [Z_1, \cdots, Z_d]$ follows a certain structural causal model $\mathcal{S}$ given as

$$Z_i \leftarrow f_i(\mathsf{Pa}(Z_i), U_i), \forall i \in \{1, \cdots, d\} \tag{35}$$

where $Z_i$ is generated from its parent variables denoted by $\mathsf{Pa}(Z_i)$ using the mechanism $f_i : \Pi_{\mathsf{Pa}(Z_i)}\mathcal{Z}_i \times \mathcal{U}_i \to \mathbb{R}$, which also takes the noise variable $U_i$ as input. The support of $Z_i$ is denoted by $\mathcal{Z}_i$ and that of $U_i$ is denoted by $\mathcal{U}_i$. Suppose we perturb $Z_k$. Under this perturbation all the latent variables for which $Z_k$ is an ancestor are going to be also affected, while keeping the rest of the variables unchanged.

Post the perturbation, the immediate children of $Z_k$ are affected and then their children and so on. Therefore, it is reasonable to assume that we first observe the impact of perturbation on $Z_k$ itself and eventually observe the impact on child variables. Consider a sample point $[(z_1, \cdots z_d), (x_1, \cdots, x_n)]$ generated by equation (1). The different observations under perturbations are

- **Pre perturbation:** $[(z_1, \cdots z_k, \cdots, z_d), (x_1, \cdots, x_n)]$
- **At the time of perturbation:** $[(z_1, \cdots z_k + \delta, \cdots, z_d), (x_1', \cdots, x_n')]$
- **Sufficiently long after the perturbation:** $[(z_1, \cdots z_k + \delta, \cdots, z_d''), (x_1'', \cdots, x_n'')]$

In the above, the latent of the sample pre perturbation and at the time of perturbation only differ in the perturbed components. However, when sufficient period has passed, other latent variables that are on the downstream path from $Z_k$ also change. In this work, we only deal with original samples and the samples at the time of perturbation. In works that rely on causal interventions such as Brehmer et al. (2022), one assumes access to samples before perturbation and those generated sufficiently long after the perturbation.

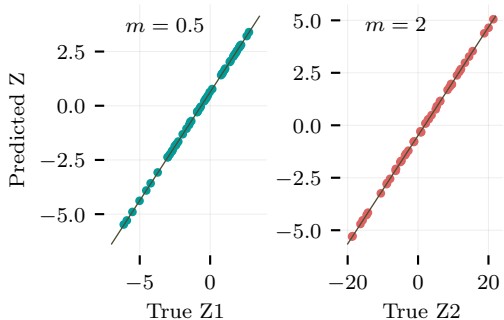

Figure 4: Regression of predicted latent values against true latent values for componentwise perturbations ($d = 10$).

### A.3 Supplementary materials for experiments

**Loss function, architecture, and other hyperparameters**    In all the experiments, we optimized equation (4) with square error loss. The encoder $f$ was an MLP with two hidden layers of size 100 for the low-dimensional synthetic experiments and a ResNet-18 (He et al., 2015) for the image-based experiments. For the low-dimensional synthetic experiments, we used the Adam optimizer (Kingma and Ba, 2014) with a learning rate of 0.005 with batches of 10000 examples for 2000 epochs. For the image-based experiments, we trained online with a learning rate of $1e - 4$ and a batch size of 100.

**Evaluation metrics**    Blockwise MCC (BMCC) is a natural extension of MCC. We compute the $R^2$ score (using linear regression) between every pair of blocks impacted under true perturbation and the guessed perturbation. We find the optimal matching between pairs of blocks to maximize the average $R^2$ score between the matched blocks. We report the $R^2$ score under the optimal matching in Table 1.

**Supplementary figures**    In Figure 4, we plot the predicted latents against the true latent value for two of the ten latent dimensions (the two dimensions that we plot are randomly selected) when we perturb one component at a time (setting corresponds to the paragraph on non-overlapping perturbations in Section 4). The plot shows a linear relationship between the true and the predicted latent; note that there are different slope and intercept for the different latents. The slope depends on the ratio between the change in the true latents and the predicted latent. In Figure 5, we plot the predicted latents against the true latent value for two of the ten latent dimensions (the two dimensions that we plot are randomly selected) when we perturb a block of two components at a time and the blocks overlap (setting corresponds to the paragraph on overlapping perturbations in Section 4). In Figure 6, we show a full set of images for the experiment shown in Figure 1.

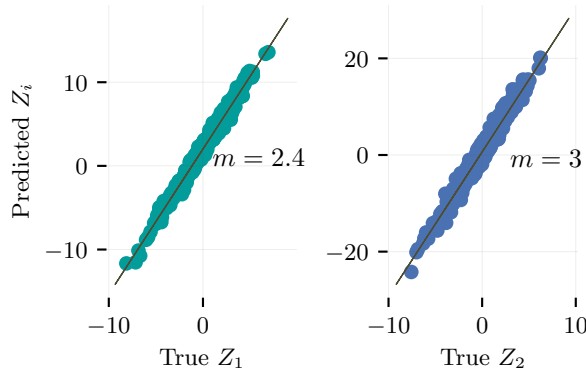

Figure 5: Regression of predicted latent values against true latent values for overlapping perturbations ($d = 10$).

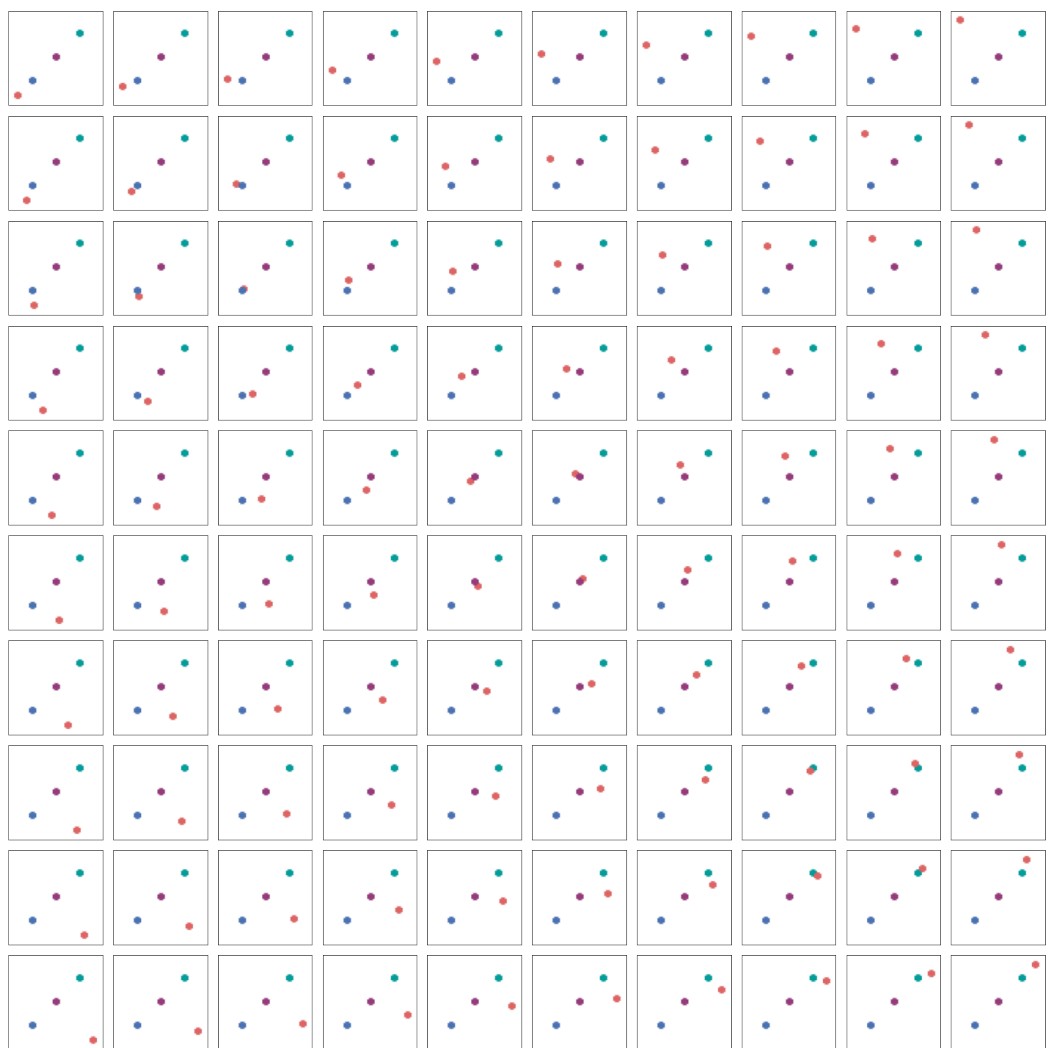

Figure 6: Full set of images for the experiment shown in Figure 1 used to render the supplementary animation. The three balls on the diagonal are stationary throughout and the fourth ball is moved across a $10 \times 10$ grid; we get the associated network predictions and animate them to show the predicted movement of the stationary balls in the attached animation.

Table 4: Comparing MCC and BMCC for stochastic perturbations

| $d$ | MCC C-wise | BMCC B-wise | MCC (overlap) C-wise |
|---|---|---|---|
| 6 | $0.99 \pm 0.00$ | $0.99 \pm 0.00$ | $0.95 \pm 0.00$ |
| 10 | $0.99 \pm 0.00$ | $0.99 \pm 0.00$ | $0.96 \pm 0.00$ |
| 20 | $0.99 \pm 0.00$ | $0.99 \pm 0.00$ | $0.98 \pm 0.00$ |

Table 5: Comparing MCC for perturbations (Normal latent)

| $d$ | MCC $\beta = 1$ | MCC $\beta = 10$ | MCC $\beta = 100$ |
|---|---|---|---|
| 6 | $0.68 \pm 0.04$ | $0.69 \pm 0.02$ | $0.72 \pm 0.02$ |
| 10 | $0.69 \pm 0.03$ | $0.69 \pm 0.02$ | $0.72 \pm 0.02$ |
| 20 | $0.70 \pm 0.02$ | $0.74 \pm 0.03$ | $0.72 \pm 0.04$ |

### A.3.1 Experiments for the stochastic model

In Section 4, we provided experiments for deterministic perturbation model. In Section A.2.3, we discussed the extension of the theory to stochastic perturbation model. In this section, we present the results for the experiments on stochastic perturbation model. We consider the same data generation process that is used in Table 1 and Table 2. We draw the latents from normal distribution used in Table 1 and Table 2. To each deterministic perturbation, we add standard normal noise (consistent with the data generation process described in Assumption 12). Instead of exactly equating the distribution in the LHS and RHS of the identity in equation (27), we take the expectation of the random variables on the LHS and RHS of equation (27). Note that when we take expectation we get the same identity that we use in equation (3). Therefore, we continue to use the loss defined in equation (4). We show the results of the experiments averaged over five trials in Table 4. These results show that the insights from the deterministic case carry over to the stochastic case as well.

### A.3.2 Supplementary experiments for comparisons with beta-VAE

In this section, we use $\beta$-VAE from (Higgins et al., 2016). We consider three different values of $\beta$ – 1, 10 and 100. We use the similar encoder architecture as in our earlier experiments for synthetic datasets, except now we have two linear heads one for the mean embedding and other for variance embedding. We use the same decoder architecture as the encoder architecture used for our earlier experiments for synthetic datasets. We use the Adam optimizer with a learning rate of 0.001. Recall that in Table 1 we used two choices for the latent distributions. For the normal distribution (which satisfies blockwise independence), we show the results in Table 5. For the uniform distribution, we show the results in Table 6.

Table 6: Comparing MCC for perturbations (Uniform latent)

| $d$ | MCC $\beta = 1$ | MCC $\beta = 10$ | MCC $\beta = 100$ |
|---|---|---|---|
| 6 | $0.63 \pm 0.03$ | $0.60 \pm 0.02$ | $0.50 \pm 0.08$ |
| 10 | $0.53 \pm 0.01$ | $0.51 \pm 0.01$ | $0.43 \pm 0.03$ |
| 20 | $0.42 \pm 0.01$ | $0.40 \pm 0.01$ | $0.36 \pm 0.01$ |

### A.3.3 Stationary point

Recall our learning objective is to minimize the objective given in Equation 4. We use a deep network, $\tilde{f}(\cdot; \theta)$ parameterized by $\theta$ as our encoder and we can rewrite Equation 4 as a loss function that depends on our choice of $\theta$ and $\Delta'$ (the learner's guess for the offsets),

$$\mathcal{L}(\theta, \Delta') = \mathbb{E}\Big[\big\|f(\tilde{x}_k; \theta) - f(x; \theta) - \delta'_k\big\|^2\Big] = \mathbb{E}\Big[\sum_j \big(f_j(\tilde{x}_k; \theta) - f_j(x; \theta) - \delta'_{jk}\big)^2\Big] \quad (36)$$

We take the partial derivative of the loss with respect to one of the parameters $\theta_i$ and obtain

$$\frac{\partial \mathcal{L}(\theta, \Delta')}{\partial \theta_i} = \mathbb{E}_{x, \tilde{x}_k}\Bigg[\sum_j \underbrace{(f_j(\tilde{x}_k; \theta) - f_j(x; \theta) - \delta'_{jk})}_{=: e_j(x, \tilde{x}_k, \theta)} \underbrace{(\frac{\partial f_j(\tilde{x}; \theta)}{\partial \theta_i} - \frac{\partial f_j(x; \theta)}{\partial \theta_i})}_{=: \phi_j(x, \tilde{x}_k, \theta)}\Bigg]$$

Suppose we learn a function $\tilde{f}$ for which $e_j(x, \tilde{x}_k, \theta)$ is independent of $x$ and $\tilde{x}$ and we denote it as $e_j(\theta)$ for all $j \in \{1, \cdots, d\}$. Under this assumption, we simplify the above expression as follows.

$$\frac{\partial \mathcal{L}(\theta, \Delta')}{\partial \theta_i} = \sum_j e_j(\theta) \mathbb{E}_{x, \tilde{x}_k}\Big[\phi_j(x, \tilde{x}_k, \theta)\Big] = \sum_j e_j(\theta) \mu_j(\theta)$$

where $\mu_j(\theta) = \mathbb{E}_{x, \tilde{x}_k}\Big[\phi_j(x, \tilde{x}_k, \theta)\Big]$. $\mu_j(\theta)$ measures the expected difference in the guessed perturbation for the component $j$ when parameter $\theta_i$ of the neural network is changed. If the impact of change in the parameter is similar on average across all the components, then $\mu_j(\theta) = \mu_k(\theta) = \mu(\theta)$ for all $j \neq k$, which leads to

$$\frac{\partial \mathcal{L}(\theta, \Delta')}{\partial \theta_i} = \sum_j e_j(\theta) \mathbb{E}_{x, \tilde{x}_k}\Big[\phi_j(x, \tilde{x}_k, \theta)\Big] = \mu(\theta) \sum_j e_j(\theta) \quad (37)$$

Under these conditions, this is a stationary point if $\sum_j e_j(\theta) = 0$. Empirically we observe that if $j$ is perturbed by $c$, then $e_j(\theta) = \frac{c}{2}$ and other components $k \neq j$, $e_j(\theta) = \frac{-c}{2(n_{\text{balls}} - 1)}$. If we substitute this in the equation above, we find that the partial derivative is zero. Since this holds for all the components $\theta_i$, we can conclude that the point observed empirically is a stationary point. Under the assumption that $e_j(x, \tilde{x}_k, \theta)$ is independent of $x, \tilde{x}_k$, we can follow the analysis presented in proof of Theorem 1, we get $\hat{z} = Az + c$. If $z$ changes by $[c, 0, \cdots, 0]$, then $\hat{z} = [\frac{c}{2}, -\frac{c}{2(n_{\text{balls}} - 1)}, \cdots, -\frac{c}{2(n_{\text{balls}} - 1)}]$. We use this to obtain $A[i, j] = \frac{-1}{2(n_{\text{balls}} - 1)}$, where $i \neq j$ and $A[i, i] = \frac{1}{2}$. If $n_{\text{balls}} = \infty$, then $A$ is a diagonal matrix, which implies that the MCC is one. In the discussion above, we assumed that the learner knows the component that changes. If the learner does not know the component that changes, then that introduces permutation errors as well.

### A.3.4 Compute used

The synthetic experiments were conducted on a 2.2 GHz Quad-core Intel Core i7. The image-based experiments were each conducted on a single GPU on a 6 core node with 16GB of allocated memory. The nodes were requested from am internal shared compute cluster with approximately 500 GPUs shared across a large number of users. Most of the GPUs are Nvidia RTX-8000 and a small number are Nvidia V100s; both types of GPUs were used to conduct the experiments depending on availability.

### A.3.5 Assets used and the license details

We used the code from https://github.com/brendel-group/cl-ica, which uses the MIT license. We also used code from https://github.com/pygame/, which is distributed under GNU LGPL version 2.1.