# OpenReview forum: "Weakly Supervised Representation Learning with Sparse Perturbations"
_NeurIPS.cc/2022/Conference — NeurIPS 2022 Accept_

### Official Review · Reviewer_zmwx · 2022-07-09

**Rating:** 6
**Confidence:** 4
**Soundness:** 4 excellent
**Presentation:** 3 good
**Contribution:** 2 fair

**Summary:**

This paper shows that if given data generated with block-wise (i.e. sparse) perturbations on the latent, then the latents are identifiable up to affine transformation on the blocks.
- In particular, in the special case of block size = 1, i.e. the perturbations change 1 coordinate at a time, then the latent is identifiable up to coordinate-wise permutation and scaling.
- For general blocks, the latent is identifiable up to affine transformations on the set of intersections and differences of the blocks (for non-overlapping blocks, this set contains the blocks themselves).

The paper also presents empirical results on synthetic data to corroborate the theory.


**Questions:**

Numbers in Table 1-3 need more contexts: how to interpret an imperfect MCC value, e.g. 0.99 vs 0.95 vs 0.85?

Side note: Perhaps considering having the same numbering for theorems in the main paper and the appendix for easier reference.

=== Update === The authors have addressed my concerns in their response.

**Limitations:**

The paper discusses limitations and future directions.

There is no direct societal impact of the work.

**Strengths And Weaknesses:**

Strengths
- The paper discusses the related work well.
- Assumption 6 (i.e. knowing the group id) is unrealistic but is relaxed in the appendix.
- The paper is well structured and easy to follow.

I have no major complaints about the paper, but I'm concerned about the novelty and significance and hope to hear the authors' comment on this.
- Novelty: It's true that this perturbation setup is different from prior work which rely on auxiliary variables, mechanisms, or assumptions on the structure of the latent. However, I find the idea of coordinate/block-wise perturbation conceptually very similar to the idea of block-wise augmentation (i.e. only act on "style") in Von Kügelgen et. al. 21.
- Significance / applicability of the results: The identifiability results requires the learner to know the type of perturbation, which is often unrealistic.
- I find using the word "sparsity" misleading, since the identifiability results are more about being block-wise, regardless of the size of the block (i.e. sparsity). Perhaps rephrasing it could make the main idea clearer.

=== Update === The authors have addressed my concerns in their response.

---

> ### Author Response · Authors · 2022-08-02
> **Thank you for the review! Please see our responses below.**
>
> We thank the reviewer for the useful feedback. We have incorporated the suggested changes. Please see the supplement to find these changes highlighted in the color blue.
>
> 1. **Regarding comparison with Von Kügelgen et al. 21**
> In Von Kügelgen et al. 21, the latent are divided into two parts -- the content block and the style block. Across augmentations, style is varied and content is fixed. Von Kügelgen et al. 21 leverage this invariance of the content across augmentations to learn the content block and not the style block. To summarize, they leverage **invariance of content latent across different views as the key signal**. In our case, the perturbations act on different blocks of latents. In stark contrast to Von Kügelgen et al. 21, we **leverage sparsity of changes, i.e., we exploit both the varying part and the invariant part** to identify all the distinct blocks and not just the content block. To give an example, suppose there are multiple balls in an image. Our theory tells us how to identify the locations of all the balls by leveraging their sparse movements, i.e., only a few balls move at a time. In contrast, Von Kügelgen et al. can identify only one ball's location by forcing its location to be fixed (treating it as the content block) and changing the position of all the other balls (treating them as the style block).
>
> 2. **Regarding novelty and significance**
>     a) We would like to emphasize that our work is the **first to show identification guarantees without making any assumptions on the latent variable distribution**  (except for the benign assumption regarding the existence of probability density over the latents) unlike existing works that make strong assumptions (conditional independence etc.).
>
>     b) Our work is the **first to establish a spectrum of identification results from full identification to partial identification** depending on the nature of sparse perturbations (ranging from blockwise to overlapping blockwise). We are also the **first to establish full identification even under overlapping perturbations**.
>
>    c) In the revision post rebuttal, we have also added two generalizations a) our **results generalize to stochastic perturbations** (See Section A.2.3), b) our **results also generalize to non-linear perturbation mechanisms** (See Section A.2.4).
>
>    d) The field of identifiable representation learning strives to find minimal assumptions that a lot of common datasets satisfy under which strong identification guarantees can be established. Our work is the first to show the power of  sparse perturbations/transitions in its full generality. We believe our work can serve as a foundation to building a range of principled methods for representation learning that rely on exploiting sparse changes.
>
>
>
> 3. **Regarding the knowledge of type of perturbation** We do not need to know the types of perturbations. In Section A.2.2 we describe how the theory extends to settings where we do not know the perturbation types.
>
> 4. **Regarding interpretation of imperfect MCC**  The source of imperfection in MCC can be due to the optimization issues, i.e., it was difficult for the optimizer to find solutions close to the global minima of the proposed loss.   Below, we clarify what it means to have a certain MCC value with the help of an example.
>
>    Suppose there are two latent variables $Z_1,Z_2$. We estimate $\hat{Z}_1, \hat{Z}_2$. Suppose
>     $\hat{Z}_1 = Z_1 + \alpha Z_2$
>     $\hat{Z}_2 = Z_2 + \alpha Z_1$
>
>
>      with $\alpha \in (0,1)$. Assume $Z_1$ and $Z_2$ are both zero mean, unit variance and are not correlated. The correlation between
>    $\hat{Z}_1$ and $Z_1$ is $\frac{1}{\sqrt{1+\alpha^2}}$. Following the definition of MCC we get that $\mathsf{MCC} = \frac{1}{\sqrt{1+\alpha^2}}$. Thus we get $\alpha =\pm \sqrt{\Big(\frac{1}{\mathsf{MCC}^2}-1\Big)}$. A simple and coarse intepretation of MCC is that it captures the extent to which an estimated latent is disentangled from the other latents. The strength of dependence on other components decreases as MCC increases and is given as  $\sqrt{\Big(\frac{1}{\mathsf{MCC}^2}-1\Big)}$.  At $\mathsf{MCC}=1$ note that $\alpha=0$, which implies no dependence on the other component.
>
> 5.  We will make other changes to theorem numbers and clarify the confusions arising from the usage of the term sparsity. We use the word sparsity to emphasize the fact that a few latents change and others are fixed. This is a consequence of the fact that the perturbation vectors are sparse.

---

> > ### Comment · Reviewer_zmwx · 2022-08-06
> > **Thank you for the clarifications**
> >
> > Thank you for the response. My concerns have been addressed; I have raised the score and recommend acceptance.

---

### Official Review · Reviewer_8xUA · 2022-07-11

**Rating:** 7
**Confidence:** 4
**Soundness:** 3 good
**Presentation:** 4 excellent
**Contribution:** 3 good

**Summary:**

This paper extends the literature on using sparse latent perturbations for identifiable representation learning. It assumes that there is a fixed set of perturbations, each of which only affects a single latent or a block of latents, in which case each block or the smallest overlap between two blocks can be identified up to affine transformations. In contrast to previous work, no independence assumptions on the latent distributions are necessary and the assumptions are a bit closer to reinforcement learning settings.

**Questions:**

* In one experiment you are selecting the perturbations randomly. What happens if you also perturb the scale of each action, so if each delta_k is scaled with a gamma that is drawn from a narrow Gaussian distribution around 1?
* At least intuitively the assumptions seem quite strong. Do you have an intuition for what assumptions could potentially be relaxed in future work?
* In the image-experiment, why do the latent factors change for all objects upon translation of one? This sounds counter-intuitive and not really in line with the training objective. Also, if this effect is robustly present, it seems to contradict the theory.

**Limitations:**

I don't see negative societal implications.

**Strengths And Weaknesses:**

The paper adds an interesting angle that is amenable to situations in which an agent controls its environment through discrete and finite actions. The manuscript is very well written, the definitions and proofs seem solid and the experiments are mostly sufficient to verify the claims (with one exception, see below). At the same time, some of the assumptions seem quite unrealistic, in particular that all perturbations need to be present for each sample, and that the set of perturbations are completely fixed. Nonetheless, the direction is interesting, in particular because it relaxes the strong assumptions on the marginal that so far had to be in place.

---

> ### Author Response · Authors · 2022-08-02
> **Thank you for the review! Please see our responses below.**
>
> We thank the reviewer for the insightful feedback. We have incorporated the suggested changes. Please see the supplement to find these changes highlighted in the color blue.
>
>  1. **Regarding stochastic perturbations:** You have asked what would happen if there was randomness in the scale of delta. We **extend our theory to incorporate stochastic perturbations** and also **provide new experiments** for the same. We contrast the deterministic model with stochastic model below.
>      >**Deterministic perturbation:** $\tilde{Z}_k\leftarrow Z + \delta_k$
>
>       >**Stochastic perturbation:** $\tilde{Z}_k\leftarrow Z + \delta_k +N_k$
>
>      where $Z$ is current latent, $\tilde{Z}_k$ is perturbed latent, $N_k$ is the noise vector.  uppose $\delta_k$ and $N_k$ are one-sparse, i.e., they perturb one component and leave the other components of the latent untouched.  Under one-sparse perturbations, we show that that estimated latent $\hat{Z}$ achieves perfect disentanglement with respect to the true latent $Z$. If one component of true $Z$ changes, then only one component of $\hat{Z}$ changes.   **Note that randomness allows different instances to have different amount of perturbation applied to them, which is something you also brought up.**
>
>       We show that the experimental insights from Table 1 and 2 in main manuscript carry over to the stochastic perturbation setting. In the table below, we carry out experiments for three different types of perturbations (componentwise, blockwise with no overlap, blockwise with overlap) with additive standard Gaussian noise. The proposed method used in the main body of the paper continues to achieve a high MCC score.
>
>  | $d$     | MCC (Component-wise)| BMCC (Blockwise no overlap)  |   MCC (Blockwise with overlap) |
>    | :---        |    :----:   |          :---:    |--: |
>    | $6$         |   $0.99 \pm 0.00$     | $0.99 \pm 0.00$        | $0.95 \pm 0.00$  |
>    |$10$         | $0.99 \pm 0.00$       | $0.99 \pm 0.00$        | $0.97 \pm 0.00$  |
>    |$20$         |    $0.99 \pm 0.00$    | $0.99 \pm 0.00$        | $0.98 \pm 0.00$  |
>
> 2. **Regarding assumptions and extensions for future work:** In the above bullet, we stated that the theory extends to stochastic perturbations. We also discuss another possible relaxation of the assumptions inspired from real world settings. In the main body, we consider fixed vectors $\delta_k$ that are sparse. In the supplement, we extend results to sparse mechanisms $m_k(Z)$. $m_k(Z)$ impacts few latents and does not change the remaining latents.
>    >**Deterministic perturbation:** $\tilde{Z}_k\leftarrow Z + \delta_k$
>
>    >**Deterministc non-linear mechanism based perturbation:** $\tilde{Z}_k\leftarrow Z+ m_k(Z)$
>
>
>    For one-sparse mechanisms, we show that our method is guaranteed to achieve perfect disentanglement.
>    For future work we believe  it would be very fruitful to develop as general a framework as possible with noisy non-linear perturbations.
>
>
> 3. **Regarding image-based experiment:** In some of our image-based experiments with overlapping blocks, we found that the motion of one ball also impacted the estimated latent of other balls. However, the latents of other balls are impacted by a factor inversely proportional to the number of balls. Therefore, this phenomenon disappears when we increase the number of balls. The theory states that if we can achieve the global minima, then the MCC score should be one and the motion of one ball should impact only the corresponding latent and not the rest. However, it is possible that the SGD-driven dynamic settles at a stationary point (e.g., local minima) and does not achieve the global minimum. Interestingly, the dynamic achieves this stationary point whose behavior mimics that of the global minimum as the number of balls increase. For further discussion on this, see the last paragraph of the experiments section and the stationary point section in the supplement.
>
> 4. **Regarding the assumption that all perturbations need to be present for each sample:** We made this assumption for ease of exposition. We can relax this in theory and experiments by considering a time series dataset. Suppose we take an image and carry out one perturbation at time one, then another perturbation at time two, and so on. For such a data generation process also, the results extend.

---

> > ### Comment · Reviewer_8xUA · 2022-08-05
> > **Thanks**
> >
> > Thanks for your thoughtful rebuttal and for addressing my questions. I have one follow-up question on point 3. Here you hypothesise that you simply find a local instead of the global minimum, but I find it very unintuitive that the local minimum is the most dense one (all latents change). One way to analyse this situation is by comparing the training losses in the following three scenarios for a relatively small set of balls:
> >
> > * Train a model as usual (which yields perturbations in all latents) and compute training loss after convergence.
> > * Take the globally optimal model and compute the training loss.
> > * Train a model starting from the globally optimal one and see whether model stays in the global optimum.

---

> > > ### Author Response · Authors · 2022-08-05
> > > **Thank you for the follow-up! (Part 1)**
> > >
> > > We thank the reviewer for the insightful follow-up question. Based on your suggestion, we provide an analysis of the training loss and the MCC for the different stationary points below.
> > > 1. **Clarification on disentanglement and an example:**
> > > There are two factors that are critical to determine the disentanglement (measured in terms of standard metrics such as mean correlation coefficient MCC) -- a) the number of latent dimensions that the estimated latent depends on, b) the extent to which the estimated latent depends on the different components.  We start with a clarifying example.
> > >
> > >    Suppose the true $Z = [Z_1, Z_2, ..Z_d]$. Assume that $\mathbb{E}[Z_i]=0$, $\mathbb{E}[Z_i^2]=1$ for all $i \in \{1, \cdots, d\}$ and $E[Z_iZ_j]=0$ for $i\not=j$. We compare two types of imperfect models--  a) entangled with one component only and b) entangled with other components, but the extent of entanglement is distributed across $d-1$ components equally.
> > >
> > >    For a) we write $\hat{Z_i}= Z_i + \beta Z_{(i+1)\mathsf{(mod)} d}$
> > >
> > >    For b) we write $\hat{Z_i} = Z_i +\sum_{j\not=i}\frac{\beta}{(d-1)}Z_j$
> > >
> > >      where $\hat{Z_i}$ is the $i^{th}$ estimated latent component and $\beta \in (0,1)$. Note that in a) if we change a latent $Z_{i}$ by one unit, then $\hat{Z_i}$  changes by one unit, and $\hat{Z_{j}}$ (where $j=i-1$)  changes by $\beta$. On the other hand, in b) if we change $Z_{i}$ by one unit, then $\hat{Z_i}$  changes by one unit, and the rest of $\hat{Z}_{j}$'s change by $\frac{\beta}{(d-1)}$.
> > >
> > >     Disentanglement aims to ensure that changes in each true latent dimension  strongly correlate to the respective estimated component they are matched with. In model a) the correlation b/w the true component $Z_i$ and the estimated component $\hat{Z}_i$ is $\frac{1}{\sqrt{1+\beta^2}}$, which is lower than the correlation between $Z_i$ and $\hat{Z_i}$ in model b) which stands at $\frac{1}{\sqrt{1+\frac{\beta^2}{d-1}}}$. Following the definition of MCC (the higher it is the better), we obtain that the MCC of model a) stands at $\frac{1}{\sqrt{1+\beta^2}}$, which is lower than the MCC of model b) $\frac{1}{\sqrt{1+\frac{\beta^2}{d-1}}}$. MCC of model b) is higher than model a) because in model b) $\hat{Z}_i$ has a stronger correlation with the underlying true latent $Z_i$ with which it is matched. Further, for model a) even though only two components are entangled, the MCC does not improve with $d$. In case b) the MCC increases with $d$. We now describe how our training objective favors stationary points similar to model b).
> > >
> > > 2. **Further details:**  Suppose that the true latent $z$ is perturbed by one unit in its $i^{th}$ component, i.e., $\tilde{z}_i \leftarrow z_i+ 1$, where $\tilde{z}_i$, $z_i$ are the $i^{th}$ component of $\tilde{z}$, $z$ respectively. Denote $\tilde{x}$ and $x$ as the corresponding renderings of $\tilde{z}$ and $z$.
> > >
> > >     We are interested in comparing the encoders $f$ at different stationary points. Suppose these encoders are imperfect, and they estimate perturbation to be $1-\beta$ instead of $1$. Define $e_k(x, \tilde{x})$ as the difference between the true perturbation and the estimated perturbation applied to latent component $k$.    In the above example, component $i$ undergoes change by one, thus we have $e_i(x, \tilde{x}) = f_i(\tilde{x}) - f_i(x) -1 = -\beta$, where $f_i$ is the $i^{th}$ component of $f$.   Also,  $e_k (x, \tilde{x}) = f_k(\tilde{x}) - f_k(x)$ for $k\not = i$.
> > >
> > >     Following the same assumptions used to derive equation (37) in the Appendix, we obtain that at the stationary point $\sum_{k }e_k(x, \tilde{x}) = 0$. Therefore, $\sum_{k\not=i}e_k (x, \tilde{x})  = \beta$. We can rewrite the objective (equation 4) that the proposed procedure minimizes as   $\min_{f} \sum_{x,\tilde{x}} \sum_{k} e_{k}(x, \tilde{x})^2$. If $f$ is chosen from a family which has a high expressivity, then we can treat the above optimization as a separable problem over each $(x, \tilde{x})$ and concern ourselves with $\min_{f}\sum_{k} e_{k}(x, \tilde{x})^2$. We consider the different imperfect encoders (stationary points) $f$, with imperfection gap $\beta$ described above, and compare them in terms of the objective $\sum_{k}e_{k}(x, \tilde{x})^2$ that we seek to minimize. We also know that all these stationary points satisfy $\sum_{k\not=i} e_{k}(x,\tilde{x})=\beta$. If we write $e_{k}(x, \tilde{x})$ as $e_k$ for simplicity, we have the following constrained optimization
> > >    $\min \sum_{k\not=i } e_k^2$ subject to $\sum_{k\not=i} e_k =\beta$.
> > >
> > >     The solution to the above problem is $e_{k} = \frac{\beta}{d-1}$ for all $k\not=i$.  If $i$ is perturbed, then $e_k = \frac{\beta}{d-1}$ for $k\not=i$ and $e_i=(1-\beta)$. We call $\hat{Z}=f(X)$. Among the different stationary points with $\beta$ gap, the training loss picks a model $\hat{Z_i} =(1-\beta) Z_i +\sum_{j\not=i}\frac{\beta}{(d-1)}Z_j$, which is similar to model b) from Point 1. As a result, as number of balls increase, we achieve MCC=1.

---

> > > > ### Author Response · Authors · 2022-08-05
> > > > **Response continued**
> > > >
> > > > 3. **Regarding comparision with globally optimal model:** The loss of the globally optimal model is zero (the perfect encoder and the perfect estimate of perturbation have to lead to a zero loss as the prediction and actual change will exactly match). Since the globally optimal model is at zero training loss, if we start the optimization at this model, there will be no updates as it is already at the lowest possible value and gradients are zero.
> > > >
> > > >
> > > > 4. To summarize, our work, like all the papers on identification in representation learning, theoretically characterizes the identification of representation for the globally optimal model. Points 1 and 2 above give an approximate analysis of what happens if the global optimum is not achieved. Currently, the literature on identifiable representation learning mostly does not analyze the behaviors outside the global optimum. We believe it is an exciting direction for future work for the community.

---

### Official Review · Reviewer_ZbtV · 2022-07-12

**Rating:** 6
**Confidence:** 2
**Soundness:** 3 good
**Presentation:** 2 fair
**Contribution:** 2 fair

**Summary:**

In this paper, the authors propose a theoretical framework on representation learning. Different from prior approaches, the authors show a different assumption and theoretically prove that sparse perturbation on latent variables is sufficient to learn the latents that follow any continuous distribution under weak supervision. The authors further discuss more general cases, and propose an estimation procedure based on the theory. Experimental results on two datasets are shown to support the claims in theory.

**Questions:**

Given the current assumption, how would your method be applied in real-world applications?

For the experiment of image-base experiments, could you consider other baseline representation learning methods as comparison?


**Ethics Review Area:**

["I don’t know"]

**Limitations:**

The authors adequately addressed the limitations and potential negative social impact of their work.

**Strengths And Weaknesses:**

$\textbf{originality}$

This manuscript provides a novel theoretical framework for representation learning which is first used in this area to the best of my knowledge. The related works are adequately cited.

$\textbf{quality}$

The authors seem to provide sound theoretical justifcation for the proposed framework. And shows experimental results to support the claim.

$\textbf{clarity}$

The paper is overall clear in presenting intuition and experimental results. The central thoery part is difficult to follow. The authors also adequately discuss the limitation of the proposed methods, and point out future direction for improvement.

$\textbf{significance}$

Lack of applications in real-world. The significance of the proposed method requires further illustration as it might be difficult to find a real-world application given the current assumptions. The applications meet the assumptions might be very specific which undermines the significance of this manuscript.

Overall, I think the paper in the current form might be interesting to the NeurIPS community.

---

> ### Author Response · Authors · 2022-08-02
> **Thank you for the review! Please see our responses below.**
>
> We thank the reviewer for the useful feedback. We have incorporated the suggestions made. Please see the supplement to find these changes highlighted in the color blue.
>
> 1. **Concern regarding assumptions and real world applications:**  We extend our theory to more general data generation processes. We have added new results -- **both theory and experiments** -- in the supplement.
>
>    i) We **extend our theory to incorporate stochastic perturbations** and **provide new experiments** for the same. To summarize, we contrast deterministic and stochastic perturbation model below.
>    >**Deterministic perturbation:** $\tilde{Z}_k\leftarrow Z + \delta_k$
>
>     >**Stochastic perturbation:** $\tilde{Z}_k\leftarrow Z + \delta_k +N_k$
>
>    where $Z$ is current latent, $\tilde{Z}_k$ is perturbed latent, $N_k$ is the noise vector.
>      Suppose $\delta_k$ and $N_k$ are one-sparse, i.e., they perturb one component and leave the other components of the latent untouched.  Under one-sparse perturbations, we show that that estimated latent $\hat{Z}$ achieves perfect disentanglement with respect to the true latent $Z$. If one component of true $Z$ changes, then only one component of $\hat{Z}$ changes.
>    We show that the experimental insights from Table 1 and 2 in main manuscript carry over to the stochastic perturbation setting. In the table below, we carry out experiments for three different types of perturbations (componentwise, blockwise with no overlap, blockwise with overlap) with additive standard Gaussian noise. The proposed method used in the main body of the paper continues to achieve a high MCC score.
>
>    | $d$     | MCC (Component-wise)| BMCC (Blockwise no overlap)  |   MCC (Blockwise with overlap) |
>    | :---        |    :----:   |          :---:    |--: |
>    | $6$         |   $0.99 \pm 0.00$     | $0.99 \pm 0.00$        | $0.95 \pm 0.00$  |
>    |$10$         | $0.99 \pm 0.00$       | $0.99 \pm 0.00$        | $0.97 \pm 0.00$  |
>    |$20$         |    $0.99 \pm 0.00$    | $0.99 \pm 0.00$        | $0.98 \pm 0.00$  |
>
>
>    ii)  We also discuss another possible relaxation of the assumptions inspired from real world settings. In the main body, we consider fixed vectors $\delta_k$ that are sparse. In the supplement, we extend results to sparse mechanisms $m_k(Z)$. $m_k(Z)$ impacts few latents and does not change the remaining latents.
>       >**Deterministic perturbation:** $\tilde{Z}_k\leftarrow Z + \delta_k$
>
>
>       >**Deterministc non-linear mechanism based perturbation:** $\tilde{Z}_k\leftarrow Z+ m_k(Z)$
>
>    For one-sparse mechanisms (that impact one latent at a time), we show that our method is guaranteed to achieve perfect disentanglement.
>
>
>    iii) In general, we can apply our objective in equation (4)  to real-world time series data (e.g., video). Our objective enforces that the change between the estimated latent as we go from one time frame to another is sparse. We currently learn fixed sparse vectors $\delta^{'}_k$ to compare consecutive frames. For more complex settings, we can learn a sparse mechanism $\delta^{'}(x, \tilde{x}_k)$.
>
>
> 2. **Regarding comparisons with other baselines:** We have added comparisons with beta-VAE (a standard approach for disentanglement). We show the results below. For the data generation process used in Table 1, where the proposed approach achieves an almost perfect MCC of one, beta-VAE achieves MCCs between 0.5-0.75.   See the table below for the performance of $\beta$-VAE.
>
>    **Uniform latents**
>      | $d$     | MCC ($\beta=1$)| MCC ($\beta=10$)  |   MCC ($\beta=100$) |
>    | :---        |    :----:   |          :---:    |--: |
>    | $6$         |   $0.63 \pm 0.03$     | $0.60 \pm 0.02$        | $0.50 \pm 0.08$  |
>    |$10$         | $0.53 \pm 0.01$       | $0.51 \pm 0.03$        | $0.43\pm 0.02$  |
>    |$20$         |    $0.42 \pm 0.01$    | $0.40 \pm 0.01$        | $0.36 \pm 0.01$  |
>
>    **Normal latents**
>      | $d$     | MCC ($\beta=1$)| MCC ($\beta=10$)  |   MCC ($\beta=100$) |
>    | :---        |    :----:   |          :---:    |--: |
>    | $6$         |   $0.68 \pm 0.04$     | $0.69 \pm 0.02$        | $0.72 \pm 0.02$  |
>    |$10$         | $0.69 \pm 0.03$       | $0.69 \pm 0.02$        | $0.72 \pm 0.02$  |
>    |$20$         |    $0.70 \pm 0.02$    | $0.74 \pm 0.03$        | $0.72 \pm 0.04$  |

---

### Official Review · Reviewer_EVcJ · 2022-07-13

**Rating:** 6
**Confidence:** 4
**Soundness:** 4 excellent
**Presentation:** 3 good
**Contribution:** 3 good

**Summary:**

The authors study how well they can recover causal structure in inferred latent variables given sparsity in perturbations.
Roughly speaking, assuming a generative process x = g(z), the authors study the effects of applying perturbations z'=z + delta leading to x'=g(z') and whether the delta can be recovered from those under what kind of conditions/assumptions.

In this work, the authors propose a hierarchy of more and more specific assumptions leading them to formalize how more specific structured can be inferred with different types of sparsity, i.e. identification up to affine transformations, blockwise identification, etc.

The paper then has 2 simple simulation examples in which they demonstrate that the theoretical properties hold.

Edit:
Increased score post rebuttal to 6.

**Questions:**

-Can the authors make statements about robustness to uncertainty/noise even in their simplified setting with exactly known number of latents? Even if the theory can't keep up now, I would be curious to see how this shakes out empirically.
- Are there any more 'appealing' experiments the authors could share so we see a less artificial example where this may work?

**Limitations:**

no concerns

**Strengths And Weaknesses:**

Strengths:
-I really enjoyed the hierarchy of assumptions and how these permit making stronger statements quite precisely about recovery of latents under unknown arbitrary latent structure. This core part of the paper seems lucid, well explained, and interesting.
It is also surprisingly simple due to the simplified setup here.
-Another strength of the paper that stands out is that it is well positioned among the current literature, with precise statements about links to contemporary papers and the detailed conditions and assumptions they share or how they differ.
-Overall the paper is also easy to read and intuitive, which is a positive result.

Weaknesses:
-There is some emphasis on literature from recent years where weakly supervised learning and disentanglement have been around a few more years in the ML community, here there is an implicit belief that after ICA the main new result is Locatello et al. What about older/other work on disentangled representations in both VAEs and RL? Is it not related just because the term causality is used less?
- The empirical setup is almost artificially toy. I have little doubt that the techniques described in this work would be applicable in various domains fitting the assumptions, but find that the experiments chosen are almost insultingly toy and hard to compare with other work.
We have a nice RL simulator from Stefan Bauer's recent work, that might be a useful testbed here in somewhat more ambitious environments. My key issue here is that the experiments are useful, but not really on the level of utility the cited work typically tackles.
- Given the simplicity of the experiments we also see that the encoder structure is simple and suffices here. I would be curious how work like this behaves under noise and uncertainty, the idealized scenario here is noiseless and assumes exact numbers of known latents.
It would be necessary to start to go beyond that very narrow scenario to study such mechanisms for real world scenarios.

---

> ### Author Response · Authors · 2022-08-02
> **Thank you for the review! Please see our responses below (part 1).**
>
> We thank the reviewer for the insightful comments. We have made the changes based on the feedback. Please see the supplement to find these changes highlighted in the color blue.
>
> 1. **Regarding literature on VAEs and RL:** This work's central focus is providing provable guarantees for representation identification. Due to space limitations, we only focused on related works that provide provable representation identification guarantees (e.g., i-VAE [1]). In the revised manuscript, we have discussed related works that the reviewer mentioned. Please see the discussion in the supplement Section A.4 (we will move these to the main manuscript in the final draft, which allows an extra page). In the new section, we cover beta-VAE [2], DIP-VAE [3], Factor-VAE [4], Annealed VAE [5]. We also cover works in reinforcement learning literature such as DARLA [6] that rely on disentanglement methods such as beta-VAE.
> 2. **Theory and experiments extend to the stochastic perturbations:** i) The data generation process in the submission used deterministic perturbations.  We **extend our theory to stochastic perturbations** (Section A.2.3) and also **provide new experiments** (Section A.3.1) for the same. To summarize, we contrast deterministic and stochastic perturbation model below.
>    >**Deterministic perturbation:** $\tilde{Z}_k\leftarrow Z + \delta_k$
>
>    >**Stochastic perturbation:** $\tilde{Z}_k\leftarrow Z + \delta_k +N_k$
>
>    where $Z$ is the current latent, $\tilde{Z}_k$ is perturbed latent, $N_k$ is the noise vector.
>    Suppose $\delta_k$ and $N_k$ are one-sparse, i.e., they perturb one component and leave the other components of the latent untouched.  Under one-sparse perturbations, we show that that estimated latent $\hat{Z}$ achieves perfect disentanglement with respect to the true latent $Z$. If one component of true $Z$ changes, then only one component of $\hat{Z}$ changes.
>
>    We show that the experimental insights from Table 1 and 2 in main manuscript carry over to the stochastic perturbation setting. In the table below, we carry out experiments for three different types of perturbations (componentwise, blockwise with no overlap, blockwise with overlap) with standard Gaussian noise added. The proposed method used in the main body of the paper continues to achieve a high MCC score as shown in the table below.
>
>
>    | $d$     | MCC (Component-wise)| BMCC (Blockwise no overlap)  |   MCC (Blockwise with overlap) |
>    | :---        |    :----:   |          :---:    |--: |
>    | $6$         |   $0.99 \pm 0.00$     | $0.99 \pm 0.00$        | $0.95 \pm 0.00$  |
>    |$10$         | $0.99 \pm 0.00$       | $0.99 \pm 0.00$        | $0.96 \pm 0.00$  |
>    |$20$         |    $0.99 \pm 0.00$    | $0.99 \pm 0.00$        | $0.98 \pm 0.00$  |
>
>
>
>
>
>    ii)   We also discuss another possible relaxation of the assumptions inspired from real world settings (Section A.2.4). In the main body, we consider fixed vectors $\delta_k$ that are sparse. In the supplement, we extend results to sparse mechanisms $m_k(Z)$. $m_k(Z)$ impacts few latents and does not change the remaining latents. We contrast the determinstic perturbation with determinstic non-linear perturbation below.
>    >**Deterministic perturbation:** $\tilde{Z}_k\leftarrow Z + \delta_k$
>
>    >**Deterministc non-linear mechanism based perturbation:**
>       $\tilde{Z}_k\leftarrow Z+ m_k(Z)$
>
>       For one-sparse mechanisms (mechanisms that impact one latent), we show that our method is guaranteed to achieve perfect disentanglement.

---

> > ### Author Response · Authors · 2022-08-02
> > **Response continued**
> >
> > 3. **Regarding more appealing experiments**  i) Our work is primarily a theory-based paper with empirical validation of the theory. Many theory-based works in the literature in the same area have also worked with illustrative experiments just like us [7]-[13].
> >
> >    ii) We agree with the reviewer that real-world dataset based experiments are important. We are working on a separate paper, which is an empirical counterpart to our theory, that aims at translating our theory to real world settings. We thank the reviewer for suggesting Causal World [14]. We agree that it is a very good candidate test environment  for our ongoing empirical paper.
> >
> >      iii) Many works on disentanglement ($\beta$-VAE and follow-ups) have also used several variations of the dSprites dataset. The dSprites dataset contains one object in the scene. In this work, we are interested in the setting with multiple objects. Our image-based dataset is a dSprites like dataset with multiple colored objects in it. We believe that experiments on this variation of dSprites dataset are insightful even if the dataset is simple.
> >
> >
> > 4. **On the assumption of knowledge of number of latent components**  Most works in the literature on theory of non-linear ICA and its generalizations make the same assumption (see i-VAE [1]). In practice, when the number of latents is not known, it can be set as a hyperparameter and can be tuned using metrics that measure proxies for disentanglement.
> >
> >
> > ### References
> >
> > [1] Khemakhem, Ilyes, et al. "Variational autoencoders and nonlinear ica: A unifying framework." International Conference on Artificial Intelligence and Statistics. PMLR, 2020.
> >
> > [2] Higgins, Irina, et al. "beta-vae: Learning basic visual concepts with a constrained variational framework." (2016).
> >
> > [3] Kumar, Abhishek, Prasanna Sattigeri, and Avinash Balakrishnan. "Variational inference of disentangled latent concepts from unlabeled observations." arXiv preprint arXiv:1711.00848 (2017).
> >
> > [4] Kim, Hyunjik, and Andriy Mnih. "Disentangling by factorising." International Conference on Machine Learning. PMLR, 2018.
> >
> > [5] Burgess, Christopher P., et al. "Understanding disentangling in $\beta $-VAE." arXiv preprint arXiv:1804.03599 (2018).
> >
> > [6] Higgins, Irina, et al. "Darla: Improving zero-shot transfer in reinforcement learning." International Conference on Machine Learning. PMLR, 2017.
> >
> > [7] Hyvarinen, Aapo, and Hiroshi Morioka. "Nonlinear ICA of temporally dependent stationary sources." Artificial Intelligence and Statistics. PMLR, 2017.
> >
> >    [8] Hyvarinen, Aapo, Hiroaki Sasaki, and Richard Turner. "Nonlinear ICA using auxiliary variables and generalized contrastive learning." The 22nd International Conference on Artificial Intelligence and Statistics. PMLR, 2019.
> >
> >    [9] Kivva, Bohdan, et al. "Identifiability of deep generative models under mixture priors without auxiliary information." arXiv preprint arXiv:2206.10044 (2022).
> >
> >    [10] Lachapelle, Sébastien, et al. "Disentanglement via mechanism sparsity regularization: A new principle for nonlinear ICA." Conference on Causal Learning and Reasoning. PMLR, 2022.
> >
> >
> >    [11] Gresele, Luigi, et al. "Independent mechanism analysis, a new concept?." Advances in neural information processing systems 34 (2021): 28233-28248.
> >
> >    [12] Gresele, Luigi, et al. "The incomplete rosetta stone problem: Identifiability results for multi-view nonlinear ica." Uncertainty in Artificial Intelligence. PMLR, 2020.
> >
> >    [13] Hälvä, Hermanni, and Aapo Hyvarinen. "Hidden Markov nonlinear ICA: Unsupervised learning from nonstationary time series." Conference on Uncertainty in Artificial Intelligence. PMLR, 2020.
> >
> >    [14] Ahmed, Ossama, et al. "Causalworld: A robotic manipulation benchmark for causal structure and transfer learning." arXiv preprint arXiv:2010.04296 (2020).

---

### Author Response · Authors · 2022-08-02
**Summary of the changes**



We thank all the reviewers for their very useful feedback. We have incorporated the reviewers' suggestions. The changes can be found highlighted in the color blue in the supplement.  Here is the summary of the key changes that we have made.

1. **Extension to stochastic perturbations**: Reviewer EVcJ and 8xUA indicated that it would be helpful to see the implications of considering stochastic perturbations instead of deterministic perturbations. We show that our theory extends to stochastic perturbations. Further, we also conduct experiments to show that the proposed method continues to be robust to stochastic perturbations, i.e., it continues to achieve a high MCC value.
2. **Extension to non-linear perturbations**: In the submission, we used fixed additive perturbation vectors. We show that our theory extends to more general perturbation models, i.e., the perturbation applied to a point $z$ depends on the value of $z$.
3. **Addition of related works on different VAEs and works in RL**: To incorporate Reviewer EVcJ's suggestion, we have added a section in the supplement titled other related works to discuss families of VAEs that were proposed to achieve disentanglement. We also discuss works in RL that explore ideas related to disentanglement.
4. **Comparison with beta-VAE:** To incorporate Reviewer ZbtV's suggestion,  we added comparisons for beta-VAE for different beta values in the supplement.

---

### Meta-Review · Area_Chair_W8yg · 2022-08-30

**Recommendation:** Accept
**Confidence:** Less certain

**Metareview:**

This submission theoretically analyzes when it is possible to identify a latent representation, up to various classes of symmetries, under the assumption that one has access to observations corresponding to sparse changes to the latent variables. This is primarily a theoretical paper, with a small amount of empirical validation. Overall, reviewers found the paper interesting and appreciated the granularity of the conclusions. They had concerns about the relationship to some of the more traditional prior work in the area as well as the realism of some of the assumptions, but subsequent discussion resolved the reviewers’ concerns. All reviewers now favor acceptance, so I believe this paper should be accepted.

**Award:**

No

---

### Decision · Program_Chairs · 2022-09-14

Accept